



# Characterizing hail-prone environments using convection-permitting reanalysis and overshooting top detections over south-central Europe

Antonio Giordani [1,2], Michael Kunz [3], Kristopher M. Bedka [4], Heinz Jürgen Punge [3], Tiziana Paccagnella [2], Valentina Pavan [2], Ines M. L. Cerenzia [2], and Silvana Di Sabatino [1]

[1]Department of Physics and Astronomy (DIFA) "Augusto Righi", University of Bologna, Bologna, Italy
[2]ARPAE-SIMC Emilia Romagna, Bologna, Italy
[3]Institute of Meteorology and Climate Research, Karlsruhe Institute of Technology (KIT), Karlsruhe, Germany
[4]NASA Langley Research Center, Science Directorate, Climate Science Branch, Hampton, VA, USA

**Correspondence:** Antonio Giordani (antonio.giordani3@unibo.it)

**Abstract.** The challenges associated with reliably observing and simulating hazardous hailstorms call for new approaches that combine information from different available sources, such as remote sensing instruments, observations, or numerical modeling, to improve understanding of where and when severe hail most often occurs. In this work, a proxy for hail frequency is developed by combining overshooting cloud top (OT) detections from the Meteosat Second Generation (MSG) weather satellite

with convection-permitting SPHERA reanalysis predictors describing hail-favorable environmental conditions. Atmospheric properties associated with ground-based reports from the European Severe Weather Database (ESWD) are considered to define specific criteria for data filtering. Five convection-related parameters from reanalysis data quantifying key ingredients for hailstorm occurrence enter the filter, namely: most unstable convective available potential energy (CAPE), K index, surface lifted index, deep-layer shear, and freezing level height. A hail frequency estimate over the extended summer season (April-October)

in south-central Europe is presented for a test period of 5 years (2016-2020). OT-derived hail frequency peaks at around 15 UTC in June-July over the pre-Alpine regions and the northern Adriatic sea. The hail proxy statistically matches with $\sim 62\%$ of confirmed ESWD reports, which is roughly 22% more than the previous estimate over Europe coupling deterministic satellite detections with coarser global reanalysis ambient conditions. The separation of hail events according to their severity highlights enhanced appropriateness of the method for large-hail-producing hailstorms (with hailstones diameters $\geq 3$ cm). Further,

signatures for small-hail missed occurrences are identified, which are characterized by lower instability and organization, and warmer cloud-top temperatures.

## 1 Introduction

Hailstorms cause billions of Euros of damage every year by severely damaging buildings (Paterson and Sankaran, 1994), crops (Zhou et al., 2016), vehicles (Hohl et al., 2002), and infrastructure. Individual hail events can produce losses exceeding 1 billion

EUR (Gunturi and Tippett, 2017), as reported for Europe (Kunz et al., 2018), the United States (Changnon and Burroughs, 2003), and Australia (Yeo et al., 1999). Hail can form in severe thunderstorms with updrafts that penetrate above their local





cirrus anvil level, containing large amounts of supercooled liquid content, and having a sufficient lifetime for the accretion process that forms graupel and hail (Knight and Knight, 2001; Houze Jr, 2014). Anthropogenic global warming is expected to increase air temperature and so its water holding capacity; as a consequence, these changes may result locally in increased low-level atmospheric moisture, leading to increased convective instability (Trapp et al., 2007; Brooks, 2013; Rasmussen et al., 2020), higher freezing level altitudes (Xie et al., 2008; Mahoney et al., 2012; Dessens et al., 2015), and possibly deviations in vertical wind shear typical values (Trapp et al., 2007; Brooks, 2013; Brimelow et al., 2017). These conditions will generally increase the probability of hail formation and the development of larger hailstones (Dessens et al., 2015; Brimelow et al., 2017; Trapp et al., 2019). However, the response of hailstorms to climate change is highly heterogeneous (Raupach et al., 2021) and still uncertain (Allen et al., 2020; Seneviratne et al., 2021), owing to the general lack of direct hail observations, the incomplete understanding of microphysical processes controlling hail formation and growth, and the insufficient level of details in operationally implemented microphysical schemes.

The low probability of hail occurrence at a certain location makes its observation still a major challenge (Prein and Holland, 2018; Allen et al., 2020). A comprehensive, standardized, and operational surface hail observing system is still missing. The existing networks are highly variable among the different countries. The atmospheric weather station networks including dedicated sensors for hail detection usually do not cover extended regions for appropriately sampling hail (except for China - Li et al., 2018b). New generations of automatic hail recorders (HARE) are in operation in some regions, for example, in parts of Switzerland (Löffler-Mang et al., 2011; Kopp et al., 2022), but unfortunately these systems are very expensive and can not be implemented over extended areas. Ground-based hailpad networks can provide homogeneous long-term records and have been deployed in several parts of the world, such as in parts of China, France, Italy, or Croatia (e.g., Changnon Jr, 1970; Giaiotti et al., 2003; Xie et al., 2008; Sánchez et al., 2009; Palencia et al., 2010; Manzato, 2012; Dessens et al., 2015). However, they cover only smaller regions and require significant resources for their maintenance. Event-based reports, collected by storm spotters, voluntary observers, or media, constitute a precious way of retrieving hail observations. This practice has been systematically adopted in the United States (Allen et al., 2015), Australia (Allen and Allen, 2016), and Europe (Dotzek et al., 2009). For the latter, Púčik et al. (2019) present the first European Severe Weather Database (ESWD) hail climatology for 1990-2018. Nevertheless, hail reports are potentially affected by spatio-temporal heterogeneity biases (e.g., higher number of reports in densely populated areas or during daytime compared to nighttime), undersampling bias of the largest hailstones, or under-reporting in case of non-damaging hailstorms (Allen et al., 2020).

To compensate for direct hail measurements limitations, proxies retrieved from remote sensing instruments have been employed to characterize hail incidence over a certain region with higher spatio-temporal homogeneity (e.g., Murillo and Homeyer, 2019; Gobbo et al., 2021; Mecikalski et al., 2021). Radar reflectivity has been used by several authors for detecting hail and estimating its size and probability of occurrence (Puskeiler et al., 2016; Nisi et al., 2020; Fluck et al., 2021). However, radar coverage is often limited to the national scale, and limitations exist in the correct inference of hailstone size (Ortega, 2018). To overcome these restricitons in the radar observations, satellite-based products have been used to characterize hail occurrence. Satellites can sample larger regions of the world with enhanced spatial homogeneity than radars (Cecil and Blankenship, 2012). Hailstorm detection methods are based on the microwave, infrared (IR) or visible spectrum measured with passive instruments




(e.g., Cecil, 2009; Melcón et al., 2016; Bang and Cecil, 2019; Laviola et al., 2020; Khlopenkov et al., 2021). Particularly, severe convective thunderstorms are detectable in the IR as local cold spot anomalies which are commonly referred to as Overshooting cloud Tops (OTs) (Adler et al., 1985). It is well known that thunderstorms presenting satellite OTs signatures have the potential

to produce a variety of hazardous weather at the surface, such as tornadoes, heavy rainfall, large hail, or wind gusts, all of which typically concentrated near OT regions (Reynolds, 1980; Brunner et al., 2007; Setvák et al., 2013; Mikuš and Mahović, 2013; Bedka and Khlopenkov, 2016; Mecikalski et al., 2021). Moreover, several studies reported the linking between large hail at the surface and OT intensity (Bedka, 2011; Punge et al., 2014; Proud, 2015; Jurković et al., 2015; Punge et al., 2017; Bedka et al., 2018; Punge et al., 2021; Wilhelm et al., 2021; Scarino et al., 2023). OTs can rapidly form and evolve within a thunderstorm;

they usually exist for less than 15 minutes (even less than 5 minutes - Elliott et al., 2012) with a maximum diameter of roughly 15 km (Fujita, 1992; Brunner et al., 2007), and with temperatures $\leq$ 215 K (Allen et al., 2020). The detection of OTs from infrared satellite imagery has been automated by Bedka et al. (2010), and subsequently refined and optimized with probabilistic approaches by Bedka and Khlopenkov (2016) and Khlopenkov et al. (2021). However, the OT proxy alone is insufficient to characterize hail occurrence because not all severe OT-generating thunderstorms produce hail on the ground (e.g., owing to

non-supportive environmental conditions or to hailstone melt during fall in case of very high freezing level heights).

To reduce the uncertainty of single-source records, potential hail proxies are often combined with hail-favoring environmental conditions, either from proximity soundings or from numerical weather forecast or climate models, used to discriminate hail from non-hail events. These are atmospheric parameters statistically associated with hailstorm formation (e.g., Johns and Doswell III, 1992; Brooks et al., 2003). This approach improves the estimation of the potential for severe thunderstorms

(Thompson et al., 2003; Hitchens and Brooks, 2014; Tippett et al., 2014) and enables to develop hail climatologies on the global (Riemann-Campe et al., 2009; Prein and Holland, 2018; Chen et al., 2020) or regional scale (Gascón et al., 2015; Púčik et al., 2017; Li et al., 2018a; Tang et al., 2019; Taszarek et al., 2021). Reanalysis datasets play a major role in this context, given the spatial homogeneity and long-term records they provide. Several studies have estimated hail hazard by coupling large-scale reanalysis/regional climate models with lightning data (Rädler et al., 2018; Battaglioli et al., 2023), surface-based reports (Prein

and Holland, 2018; Torralba et al., 2023), or combinations of lightning, insurance loss data, severe weather reports (Mohr et al., 2015), and radar data (Taszarek et al., 2020). The combination of hail-favoring ambient conditions from reanalysis with OT detections has been applied to describe hail hazard over Europe (Punge et al., 2017), Australia (Bedka et al., 2018), and South Africa (Punge et al., 2023). However, the reanalysis-based OT filter developed in these studies rely on global datasets such as ERA-Interim (Dee et al., 2011) or ERA5 (Hersbach et al., 2020), characterized by coarse horizontal resolutions (i.e., about

80 and 25 km, respectively), which could produce significant inaccuracies. Indeed, a fine spatio-temporal resolution in the models constitutes a crucial necessity for improving the representation of deep moist convection including hail (Wilhelmson and Wicker, 2001; Bryan et al., 2003; Wu and Arakawa, 2014; Clark et al., 2016; Allen et al., 2020; Raupach et al., 2021).

Convection-permitting (CP) simulations, with horizontal grid spacings of a few km that allow to switch off physical parameterizations for convection in the models, have been demonstrated to enhance the model's skills to forecast convective

phenomena thanks to the explicit representation of most convective motions (e.g., Prein et al., 2015; Trapp and Hoogewind, 2016; Hoogewind et al., 2017; Prein et al., 2017; Liu et al., 2017; Trapp et al., 2019; Lupo et al., 2020; Giovannini et al.,



2021; Chen et al., 2021; Tiesi et al., 2022). Particularly, CP models have provided added value for the representation of tornadic or large-hail environments (Clark et al., 2013; Adams-Selin and Ziegler, 2016; Labriola et al., 2019; Gagne II et al., 2019; Manzato et al., 2020; Malečić et al., 2022). The need for an enhanced description of high-impact, rapidly-evolving and

sharply-localized atmospheric phenomena prompted the development of a new CP regional reanalysis over south-central Europe: SPHERA (High rEsolution ReAnalysis over Italy - Cerenzia et al., 2022; Giordani et al., 2023). Based on dynamical downscaling of ERA5 and driven by a non-hydrostatic model COSMO (Schättler et al., 2018) at 0.02° grid spacing, SPHERA has shown to enhance the description of weather extremes such as severe precipitation over Italy and neighboring countries (Giordani et al., 2023). This region is particularly relevant as it represents one of the major hail hotspots over Europe (i.e.,

northern Italy, Taszarek et al., 2020).

    This work aims at presenting a new method for hail hazard assessment over south-central Europe obtained by coupling satellite OT detections from Meteosat Second Generation (MSG) SEVIRI instrument with a set of environmental predictors from SPHERA and ESWD hail reports. The purpose is to obtain a new proxy for hail by optimally tuning the combination among the different data sources, and to evaluate its reliability over a 5-year test period (2016-2020) during the extended

summer season (April-October). Additionally, the high-resolution CP regional reanalysis allows a detailed representation of the environmental conditions occurring during hailstorms. Hence, in order to investigate the atmospheric states potentially associated with hail, the distributions of the numerical convective indices considered are conditionally analyzed depending on hail severity and on the ability of detection of the proxy.

    Sect. 2 presents the datasets used, while the filtering procedure developed to retain potentially hail-related OTs is described

in Sect. 3. The resulting hail frequency and the associated ambient conditions are reported in Sect. 4, while discussions and conclusions are drawn in Sect. 5.

## 2   Data

This section describes the sets of hail reports, OT detections and reanalysis proxies considered. The investigation period pertains to the extended summer (April-October), representing the climatological season for hail at mid-latitudes, over the period 2016-

2020. The reference area pertains to the entire SPHERA reanalysis domain (i.e., approximately 35-49°N; 6-19°E, Fig. 2a), including the countries of Italy, Switzerland, Austria, Slovenia, Croatia, parts of Bosnia-Herzegovina and Germany.

### 2.1   ESWD hail reports

The ESWD (https://www.eswd.eu/; Dotzek et al., 2009) constitutes a primary source for severe convective storm data in Europe and represents the largest and only multinational European hail report archive. Maintained by the European Severe Storm

Laboratory (ESSL), the ESWD provides quality-checked data collected by networks of voluntary observers, meteorological services, weather enthusiasts, and news and media reports. Thanks to technological innovations and increasing public awareness of extreme meteorological events, reports have rapidly increased in number in recent years (Groenemeijer et al., 2017). Despite recent advances, the ESWD still suffers from deficits in data representativity owing to the spatial-inhomogeneity in the





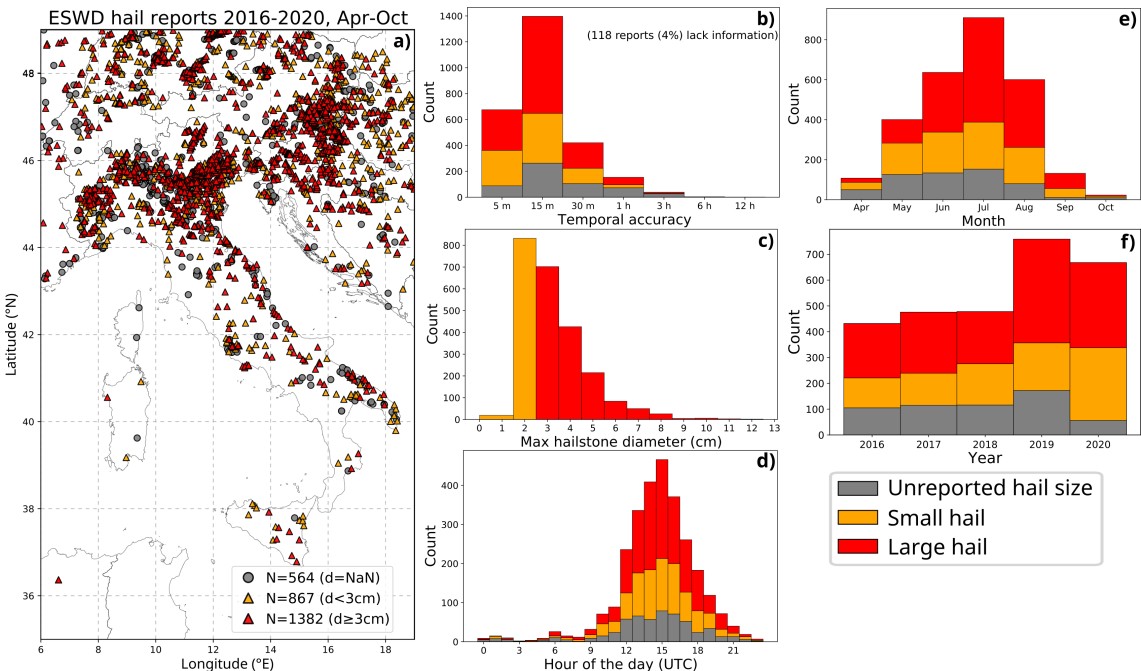

**Figure 1.** ESWD hail reports in April-October over 2016-2020. The reports are classified by distinguishing among three classes: reports with no information on hail size (in gray), small hail (maximum diameter <3 cm, in orange), and large hail (maximum diameter ⩾ 3 cm in red). a) spatial distribution, b) temporal accuracy distribution, c) maximum hailstone diameter distribution, d) number of reports per hour of the day (UTC), e) number of reports per month, and f) number of reports per year.

localization of a large part of reports, biased towards the most populated areas (i.e., the main urban centers) and the relatively

short temporal series covering only the last few years. However, the ESWD forms the only reliable source of direct hail data for Europe, including information such as location, date, hour (with an estimate of the temporal accuracy), and maximum size of hailstones. Further, an operational quality control procedure categorizes each report with different quality levels: QC0 ("as received"), QC0+ ("plausibility checked"), QC1 ("confirmed by reliable source"), and QC2 ("scientific case study").

A total of 2,813 hail reports with a minimum quality level of QC0+ were available for this study (Fig. 1), of which 2,249

(80%) contained hail size information (for a minimum hailstone diameter of 2 cm - Fig. 1c), used to quantify hail severity. Hereafter we refer to small hail reports for maximum hailstone diameters of <3 cm, large hail for reports with maximum hailstone diameters of ⩾3 cm, and very large hail for reports with maximum hailstone diameters of ⩾5 cm. These different classes account for 39, 61, and 16%, respectively, of ESWD reports with information about hail size. Their spatial distribution (Fig. 1a) shows a strong inhomogeneity with a larger density of reports in northern Italy, south-eastern Austria, eastern Slovenia,

and northern Croatia. The regions with the lowest hail reporting are central-southern Italy and all islands, the main Alpine crest (extending along the northern Italian border with France, Switzerland and Austria), and southern Balkans (southern Croatia and Bosnia and Herzegovina). 98% of the reports have a temporal accuracy of ⩽ 1h (Fig. 1b). Their temporal distribution indicates



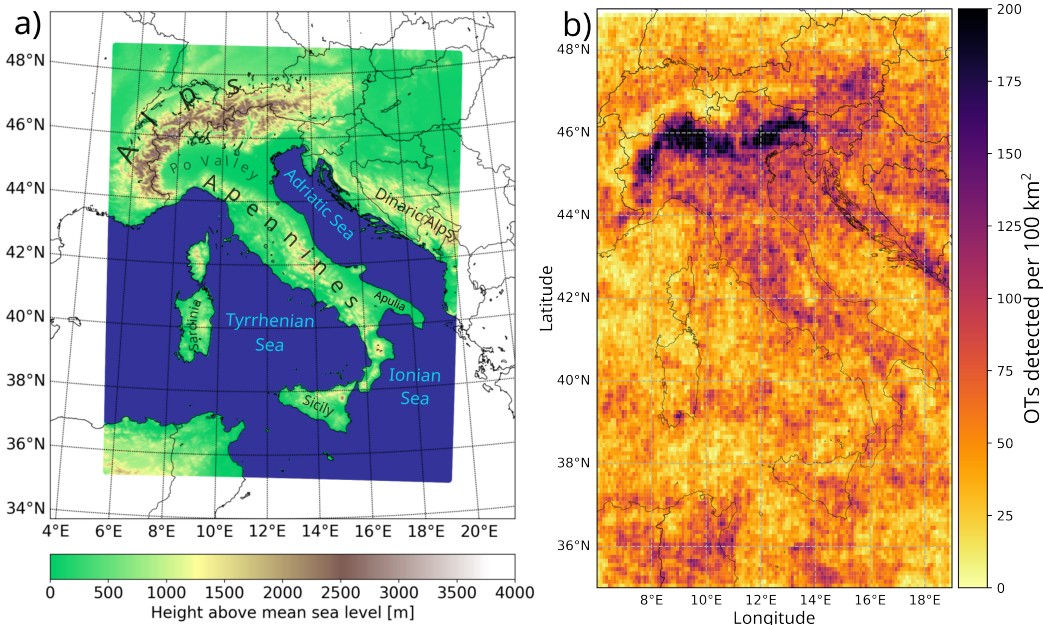

**Figure 2.** a) The spatial domain and model orography of SPHERA reanalysis. b) Number of overshooting tops detected per grid cell (on a 10-km regular grid) during April-October in 2016-2020.

the maximum probability for hail at 15 UTC (Fig. 1d) and in July (Fig. 1e), with a similar number of reports (between 400 and 500) for the first three years considered (Fig. 1f). A substantial increase is noted for 2019 and, to a slightly lesser extent, for 2020.

## 2.2 Overshooting top detections

The remote detection of OTs automated by Bedka et al. (2010) has been previously used to characterize OTs climatological distribution in North America (Bedka et al., 2010), Europe (Bedka, 2011), and Australia (Bedka et al., 2018). The OT detection algorithm relies on the comparison between clusters of cold pixels likely related to strong updrafts with a tropopause temperature, as well as with pixels consistent with the temperature of the anvil of the thunderstorm, as detected with IR satellite scans. A large temperature difference (> 6 K) helps to separate actually occurred OTs from other non-convective clouds (e.g., cirrus) as this difference is indicative of updraft penetration through the anvil of at least 1-2 km (Griffin et al., 2016). Recently, the automatic OT detection algorithm has been substantially improved by considering a probabilistic approach (Bedka and Khlopenkov, 2016; Khlopenkov et al., 2021) instead of the binary yes/no decisions based on predefined fixed temperature thresholds. The statistical combination of tropopause-relative IR brightness temperature, the prominence of a candidate OT relative to the surrounding anvil, and the spatial uniformity and size of the area covered by the anvil delivers a 3-km gridded probabilistic OT estimate across the domain. The validation of this methodology (Khlopenkov et al., 2021; Cooney et al., 2021) revealed important improvements compared to the original detection algorithm of Bedka et al. (2010).





The present study considers IR imagery from geostationary MSG Spinning Enhanced Visible and InfraRed Imager (SEVIRI)
(Schmetz et al., 2002) between 2016 to 2020 at a continuous temporal resolution of 15 minutes over south-central Europe.
Only OTs detected with the Khlopenkov et al. (2021) algorithm having a probability >50% are considered, similar to Punge
et al. (2023). This statistical constraint was derived by the comparison of OT detections with radar echo tops (Cooney et al.,
2021) that demonstrated enhanced reliability being indicative of colder and more prominent anvil-relative tops. The spatial
distribution of the 991,042 OTs detected over 872 days is shown on a 10-km regular grid in Fig. 2b. A generally higher number
of OTs over land is observed, especially around the Alps, Apennines, and Dinaric Alps mountainous ranges. The main OT
hotspot over the study domain extends throughout the northern Po valley region adjacent to the Alps, which is bounded to the
north by the most prominent minimum located in proximity of the Alpine crest (i.e., along the northern Italian border with
south-eastern France, southern Switzerland, and western Austria, Fig. 2a).

## 2.3  SPHERA reanalysis ambient predictors

The atmospheric conditions associated with convective environments that favor hail formation are described with the high-
resolution regional reanalysis SPHERA (Cerenzia et al., 2022; Giordani et al., 2023). SPHERA is a dynamical downscaling of
the global reanalysis ERA5 driven by the non-hydrostatic limited-area model COSMO at the convection-permitting horizontal
grid spacing of 0.02° over 65 vertical levels. The assimilation of regional observations (wind speed, pressure, air humidity
and temperature), coming from various sources (surface weather stations, radiosoundings, radar and aircraft reports), with
a continuous nudging scheme steers the simulations towards the observed state. Three-dimensional output is produced on a
hourly basis for south-central Europe (Fig. 2a) during 1995-2020.

The meteorological parameters selected to describe ambient conditions favorable to hailstorm development and to identify
potential hail-related OTs rely on statistical relationships between hail observations and proximal atmospheric soundings (e.g.,
Kunz, 2007; Prein and Holland, 2018; Kunz et al., 2020; Allen et al., 2020; Jelić et al., 2020). These parameters represent
the key dynamical and thermodynamical ingredients necessary for hailstorm formation: atmospheric instability and low-level
moisture, storm organization, and freezing level altitude.

Atmospheric instability and related updraft strength of a thunderstorm are particularly relevant as strong updrafts are nec-
essary for hail growth. Numerous radiosounding-based instability indices have been proposed to predict the potential for
thunderstorm development. As of today, the preference on which index is most suited to represent favorable conditions for
hail occurrence is not univocal, and could change depending on the different ambient conditions over particular regions (e.g.,
Europe compared to the United States, where the majority of the indices have been designed - Brooks, 2009; Taszarek et al.,
2020, 2021). Hence, to better identify hail-favorable meteorological conditions and reduce possible misrepresentation by using
a single quantity (that may potentially be unsuitable in some situations), three different thermodynamic parameters are con-
sidered. These are Most Unstable (MU) CAPE, K index, and Surface Lifted Index (SLI), which reported the highest skill for
severe thunderstorms predictions in central Europe (Kunz, 2007). Their different formulations (reported in Appendix A) imply
the contribution from distinct parts of the numerical model equations involved in their computation. K index relies only on the
environmental characteristics of the vertical temperature and moisture content of the atmospheric profile. On the other hand,



SLI and CAPE consider the temperature difference between the environment and the lifted parcel rising with the convective air mass, but while SLI is a "two-level" index (based on the temperature difference between the ambient and the rising parcel at 500 hPa), CAPE is an integrated measure of buoyancy over the entire vertical column.

Additionally, organized thunderstorms in the form of multicells, supercells, or mesoscale convective systems (MCS) are more likely to produce hail. Hence, the 0-6 km wind vector difference (deep layer shear - DLS), frequently used for investigating hail-favoring conditions (e.g., Trapp et al., 2007; Wellmann et al., 2020), is also included in the filter.

Finally, the amount of moisture available below the freezing level has an influence on the hydrometeor density in the updraft, and hence potentially on the growth rate of the hailstones (Johnson and Sugden, 2014; Allen et al., 2015). On the one hand, too low a freezing level may limit the amount of supercooled water in the updraft necessary for hail growth (Prein and Holland, 2018). On the other hand, thunderstorms with a high freezing level ($H_0$) are less likely to produce hail on the ground owing to enhanced melting during hailfall (Dessens et al., 2015). This causes, for example, the lower hail probability observed near the tropics where the surface atmospheric layers are generally warmer and the tropopause higher (Prein and Holland, 2018). Hence, $H_0$, defined as the altitude of the 0°C isotherm above mean sea level, is included in the filter.

While CAPE, SLI, and $H_0$ are direct outputs of SPHERA, DLS and K index are computed from temperature and wind profiles. Every parameter is available at hourly frequency at the native horizontal resolution of 0.02° (i.e., ∼2.2 km). However, local rapidly evolving deep convective processes are characterized by a low intrisic predictability and this may affect the representativity of the local indices considered. Hence, SPHERA fields are remapped to a common grid of 10 km to avoid possibly "noisy" estimates and to reduce data representativity issues.

## 3 OT-reanalysis filter design

Recent findings suggest links between convective storm severity and specific characteristics of the OT detections, such as their spatial extension (Marion et al., 2019) or the temperature gradient between the OT and the tropopause (Khlopenkov et al., 2021). However, some OTs with intense updrafts reaching the tropopause and penetrating the lower stratosphere may be associated with convective environments not necessarily supportive of severe weather phenomena such as hail. The necessary discrimination between hail- and non-hail-producing OTs can be attained by additionally considering convection-related environmental conditions estimated with reanalysis (Punge et al., 2017; Bedka et al., 2018; Punge et al., 2023).

SPHERA predictors are extracted around each OT detection in a spatio-temporal neighborhood of 0.63° x 0.63° (approximately 70 km x 70 km) over the three hours preceding an OT and the hour at which the OT is issued. This relatively large spatial matching window is required owing to the extremely localized and rapidly-evolving nature of hailstorms in order to limit double-penalty issues due to the difficulties of the models to predict the exact localization of convective processes (Ebert, 2008; Marsigli et al., 2021). Additionally, to take into account pre-convective conditions from SPHERA, a temporal window before the OT event is considered. Within this spatio-temporal neighborhood, the maximum (for CAPE, K index, and DLS) or minimum (for SLI and $H_0$) values of these parameters are extracted.





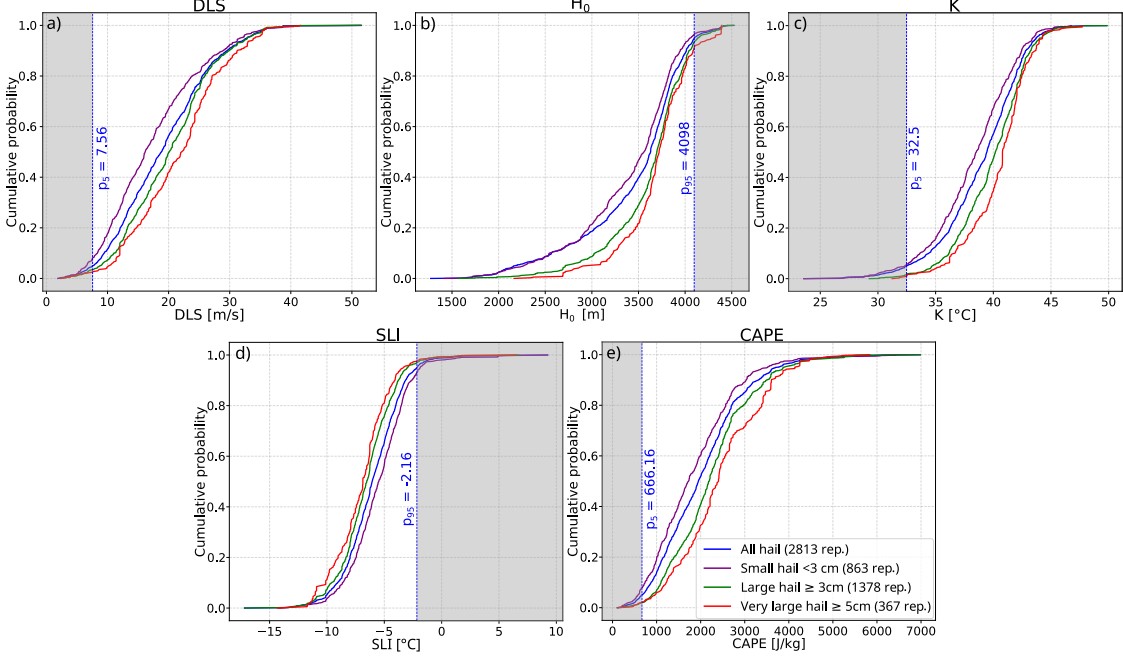

**Figure 3.** Cumulative density functions of the five parameters selected from SPHERA in the presence of ESWD hail reports. Hail reports are divided into different classes: all reports (blue lines), small hail (<3 cm, purple lines), large hail ($\geqslant$3 cm, green lines), and very large hail ($\geqslant$5 cm, red lines). The blue dashed vertical lines indicate the thresholds selected for defining the OT hail filter reported in Table 1. The shadowed portion of the distributions indicates the range of values where the filter is effective. a) DLS, b) $H_0$, c) K index, d) SLI, and e) CAPE.

The filter to select potential hail-related OTs is then constructed by employing confirmed ESWD hail reports. The environmental parameters are selected in the vicinity of the reports by considering the same spatio-temporal neighborhood used for the OTs (i.e., $0.63° \times 0.63°$ spatial window, 0-3 hours temporal window), and the thresholds are defined as percentiles $p_{th}$ of the distributions of the parameters (following the approach of Punge et al., 2017). Conversely to the latter study, where thresholds based on the 2nd or 98th percentiles were prescribed, here the slightly more stringent 5th percentile (for CAPE, K index, and DLS) or 95th percentile (for SLI and $H_0$) is selected. This is justified by the higher spatio-temporal resolution of SPHERA reanalysis (2.2 km - 1 h), which, compared to ERA-Interim (80 km - 6 h) considered by Punge et al. (2017) is expected to significantly enhance the representation of the atmospheric conditions described by the indices (e.g., in the form of sharper peaks in the parameter distributions owing to clearer distinction of the modeled dynamical features).

The ESWD-based cumulative density functions (CDFs) of the predictors are reported in Fig. 3. To investigate the relationship between each parameter and hailstorm severity, the CDFs are shown for the distribution of the entire hail reports set, and separated between small, large, and very large hail. A general shift of the predictors towards severe-convective environments is detected for increasing hail sizes. Indeed, moving from the purple to the red lines in Fig. 3, increased instability (greater CAPE and K index and lower SLI), enhanced organization (greater DLS), and higher freezing levels are noted. This suggests



**Table 1.** Variables and thresholds used in the OT filter, relative number and fraction of OTs filtered, and number of days with active OT filtering (with fractions expressed out of the 872 days when at least one OT is detected).

| Variable | Threshold | OTs filtered | Fraction | Days w. active filter |
|---:|---|---:|---:|---|
| SLI | $< p_{95} = -2.16$ °C | 111,042 | 11.2% | 679 (78%) |
| CAPE | $> p_5 = 666.16$ J kg$^{-1}$ | 98,051 | 9.9% | 669 (77%) |
| K index | $> p_5 = 32.5$ °C | 91,794 | 9.3% | 583 (67%) |
| DLS | $> p_5 = 7.56$ m s$^{-1}$ | 88,796 | 8.6% | 414 (48%) |
| $H_0$ | $< p_{95} = 4098$ m | 69,347 | 7.0% | 244 (28%) |
| Full filter | All those above | 267,900 | 27.0% | 824 (95%) |

the ability of the numerical proxies to identify hail-related ambient conditions. The shaded areas in Fig. 3 indicate the tail of
the CDFs (corresponding to the 5th or 95th percentile portions) where the filter is active.

### 3.1 Reanalysis parameters contribution to OT filtering

The thresholds obtained, the numbers of OTs and the relative fractions filtered by the full filter (applying the five conditions together) and for each parameter are listed in Table 1. Singular parameter contributions to the filter vary from 7.0 to 11.2%. Since the same OTs are sometimes filtered by more than one variable, the fraction of removed OTs with the full filter is lower
than the sum of the singular filters and reaches 27.0%. The fraction of days when instability-index filters (SLI, CAPE, and K index) are active amount to $\sim 70$ % and beyond. This suggests their dominant contribution in the OT selection compared to DLS, which filters in roughly half of the days, or to $H_0$, being active in less than one third of the days. The resulting full filter is active in almost the totality (95%) of days with at least one detected OT.

To understand the impacts of the different parameters in the OT filtering, their spatio-temporal contributions are investigated.
Figure 4 shows the spatially-distributed filtered fractions of OTs for the single-parameters filters (Fig. 4a-e) and for the full filter (Fig. 4f). Instability parameters (Fig. 4a-c) filter mainly over the sea (especially in the southern and western Mediterranean) and the Alpine crest, particularly along the Italian-Swiss border. The largest contribution over the sea is given by K index, while over land CAPE and SLI are more active. This is presumably owing to the explicit inclusion of the water vapor content in the atmospheric column in the K index, that weights more over the sea. $H_0$ (Fig. 4d) filters most OTs over lower latitudes (>60%
in Tunisia and Algeria) and high-elevation terrains, especially over the whole Alpine crest, where almost 100% of all OTs are filtered out. This enhanced removal is attributable to the generally colder atmospheric profiles found over the Alps (compared to lower-elevation regions) where the simulated topography reaches elevations as high as 3950 m in SPHERA reanalysis (Fig. 2a), and the freezing level usually exceeds the imposed threshold of 4098 m. However, as seen by the spatial distributions of ESWD reports (Fig. 1a) and OTs (Fig. 2b), the Alpine crest is the least populated region of the domain in terms of hail
reports and prominent OTs. This is attributed to the difficulties for deep-organized convective systems to develop in extremely complex terrains, in agreement with a recent climatology of lightning flashes and associated conditions for convective initiation over the Alpine area (Manzato et al., 2022b). Hence, it is believed that the limitation imposed by choice of the $H_0$ threshold is



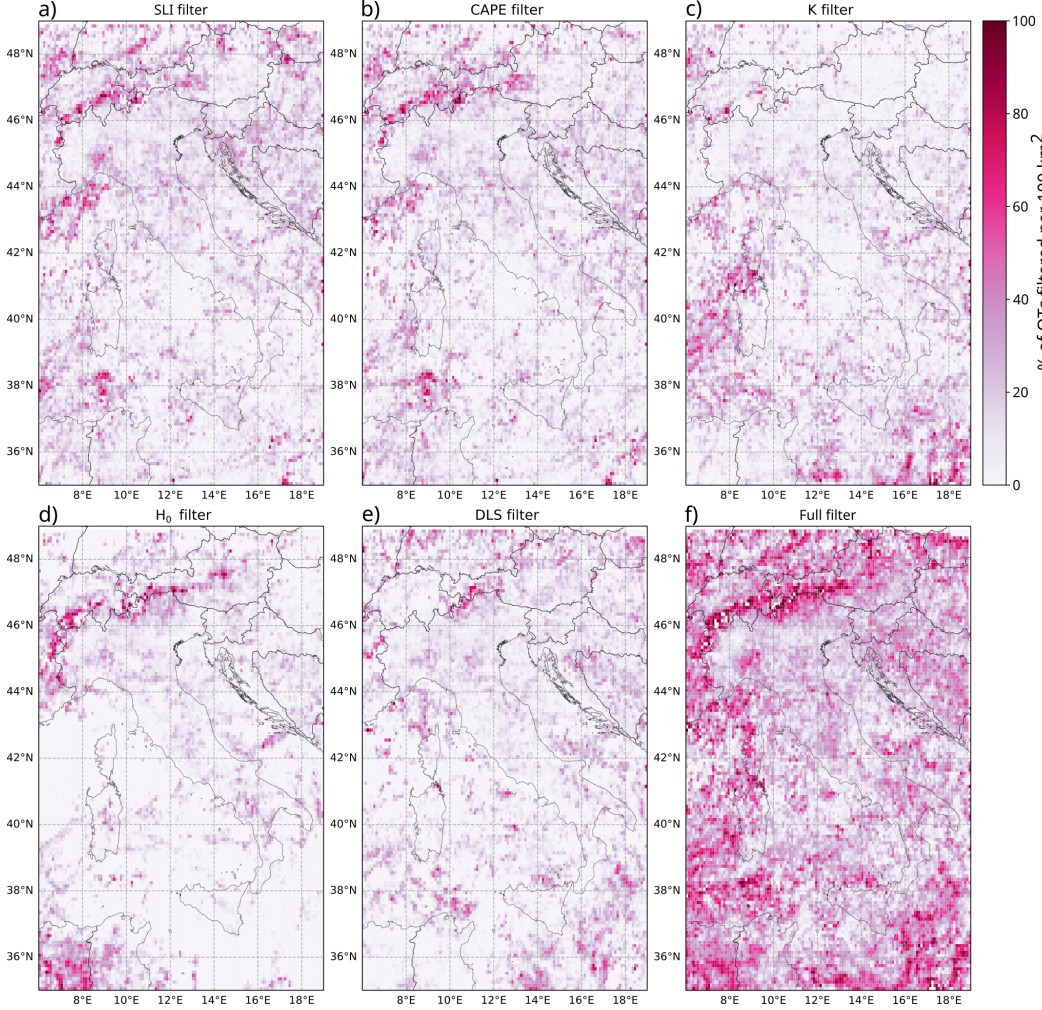

**Figure 4.** Percentage of filtered OTs per grid cell (on a 10-km regular grid). a) SLI, b) CAPE, c) K index, d) $H_0$, e) DLS, and f) full filter.

not detrimental to the analysis presented here (a possible proposal for a more sophisticated $H_0$-filtering could be topography-dependent).

The DLS filter (Fig. 4e) shows less prominent spatial peaks than other parameters, but enhanced activity in the northern part of the domain (i.e., southern Germany and northern Austria) and in the south-eastern Mediterranean sea.

     The combination of the five individual filters (Fig. 4f) delivers maximum filtering ($\sim$100%) along the northern Italian border where the main mountain peaks of the Alps are located, and substantially high ($\sim$60%) but locally variable filtering over the western and southern Mediterranean sea, southern Germany and eastern Austria. In contrast, the regions with the lowest OT

removal (<20%) are the Po valley in northern Italy, the northern Adriatic and Thyrrenean seas, and the associated Italian and Croatian coastlines.



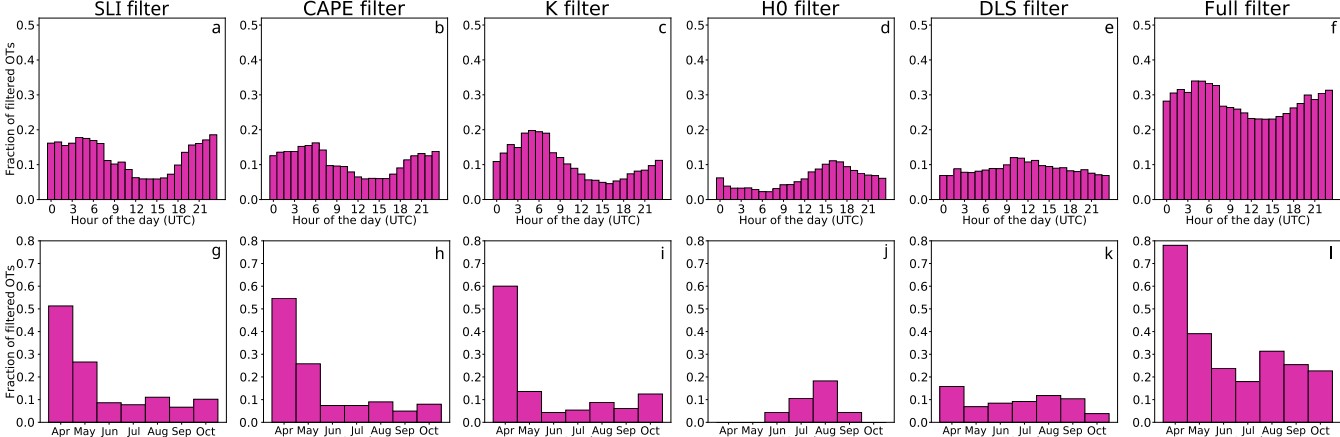

**Figure 5.** Percentage of filtered OTs aggregated over the spatial domain per hour of the day (top row) and per month (bottom row) considering singular parameters filters: a) & g) SLI, b) & h) CAPE, c) & i) K index, d) & j) $H_0$, e) & k) DLS, and f) & l) the full filter.

Figure 5 shows the fractions of filtered OTs for the different parameters depending on the hour of the day (upper row) and the month of the year (lower row), aggregated over the whole spatial domain. It should be kept in mind that the local time zone over the considered region, the Central European Summer Time (CEST), is two hours ahead of the UTC time zone. Instability parameters (SLI, CAPE, and K index) filter mainly during the night and early morning (reaching 20% around 4-7 UTC), and in April (exceeding 50%), May, and October. This reflects the lower likelihood of hail-favoring convective conditions at these times of the day and year. Conversely, their contribution to the filter is minimal when the increased heating of the boundary layer enhances the potential for convective activity and reduces convective inhibition, increasing the possibility for hail formation (e.g., Markowski and Richardson, 2011), i.e., in the central hours of the day (with less than 10% removal around 14-16 UTC) and during JJA (June July August). No evident differences among the three parameters are detected.

The $H_0$ contribution (Fig. 5d-j) is roughly opposite to that of the instability. The largest removal is found in the afternoon (∼12% at 16-18 UTC) and in late summer, especially in August (about 20%). This seasonal variation is likely linked to the warming of the lower troposphere peaking in August in this region, owing to the annual cycle of solar insolation, producing an upward shift of the freezing level. On the other hand, the daily cycle in $H_0$ filter cannot be generally related to the diurnal cycle of boundary layer warming. In fact, at altitudes of ∼4 km above sea level (a.s.l.), temperature changes are mainly driven by horizontal advective processes, rather than by vertical sensible heat fluxes, which are little affected by low-level daily variability. The largest fraction of $H_0$-driven OT removal is found over the main Alpine crest (Fig. 4d), where the atmospheric boundary layer could extend over 4 km a.s.l., despite being very shallow, implying a possible diurnal impact on the $H_0$ variation. Anyhow, as previously mentioned, for the purpose of the analysis here presented, this does not constitute a critical issue as indicated by the almost complete lack of multiple hail proxy signals in this region (i.e., hail reports, OT and lightning detections).



The DLS filter (Fig. 5e-k) shows the least diurnal and seasonal variations, with slightly higher OT removal rates around 10 UTC (∼12%) and in April (∼15%). The reduced variability in DLS filtering compared to all the other parameters is most likely attributed to its kinetic (rather than thermodynamic) nature, and to its direct relationship with synoptic-scale forcings. Indeed,

during the considered warm season of the year, the typical synoptic conditions found in the study region are dominated by a persistent anticyclonic ridge. This large-scale forcing produces a general less variable and lower wind magnitude difference between the surface and at ∼6 km altitude compared to its cold season counterpart, which is characterized by more dynamism (e.g., stronger jet streams), and associated with the DLS climatological maximum (Taszarek et al., 2018).

The resulting full filter on the daily term (Fig. 5f) shows maximum removal of more than 30% in the morning and evening

(4-7 and 20-23 UTC respectively) and to a minimum of ∼23% around 13-15 UTC. Considering the seasonal cycle (Fig. 5l), the parameters combination shows enhanced filtering in spring (April with almost 80% and May with almost 40%) followed by August with a removal above 30%. Hail-favoring conditions are most likely to be estabilished in July and June, where minimum OTs removal of ∼13% and ∼22% are issued, respectively. This tendency is in good accordance with the observed hailstorms distribution over the years considered (Fig. 1e), and with more robust 28-year ESWD-based hail climatology (Púčik

et al., 2019).

## 4 Hail frequency and ambient conditions

### 4.1 Spatio-temporal characterization

Figure 6a shows the spatial distribution of the 723,142 OTs retained after applying the filtering described in the previous section over the five extended warm seasons considered. Compared to the original distribution (Fig. 2b), a decrease in the number of

OTs over the main Alpine crest is evident, associated with the maximum removal rate in that area (Fig. 4f). Fewer OTs are also detected over land at lower latitudes (Algeria and Tunisia), over the Mediterranean sea, throughout the Apennines, and in north-eastern continental areas (Austria, Slovenia, Croatia, and Bosnia). The main hotspot of OT frequency in the region along the southern pre-Alps and northern Po valley is well preserved after filtering. Further, the resulting contrast with the minimum OT frequency found over the main Alpine crest is more pronounced than before filtering. This suggests the identification of

preferential areas for hail formation, which show good agreement with findings from Punge et al. (2017) and recent radar-based hail climatology (Nisi et al., 2020).

Hail frequency in a certain area is usually estimated as the number of hail days per year rather than counting every single hailstorm (Punge and Kunz, 2016). In this case, a potential hail day (PHD) is defined as a day when at least one hail-related OT is detected per reference area of 10 x 10 km$^2$. The sensitivity tests performed by increasing the number of OTs defining

a PHD showed stable and mutually consistent spatial structures (only from > 10 OTs per day the distributions started to lose too much detail). The resulting average PHD distribution is reported in Fig. 6b after spatial smoothing with a Gaussian filter. This is done to minimize potential uncertainties arising from spatio-temporal shifts between the OT proxy and the occurrence of hail on the ground, and to homogenize the gridded distribution. The result suggests a maximum hailstorm frequency of ≥ 7 PHDs per year in proximity of the southern Alpine slopes and ∼ 0 PHDs over the Alpine crest.



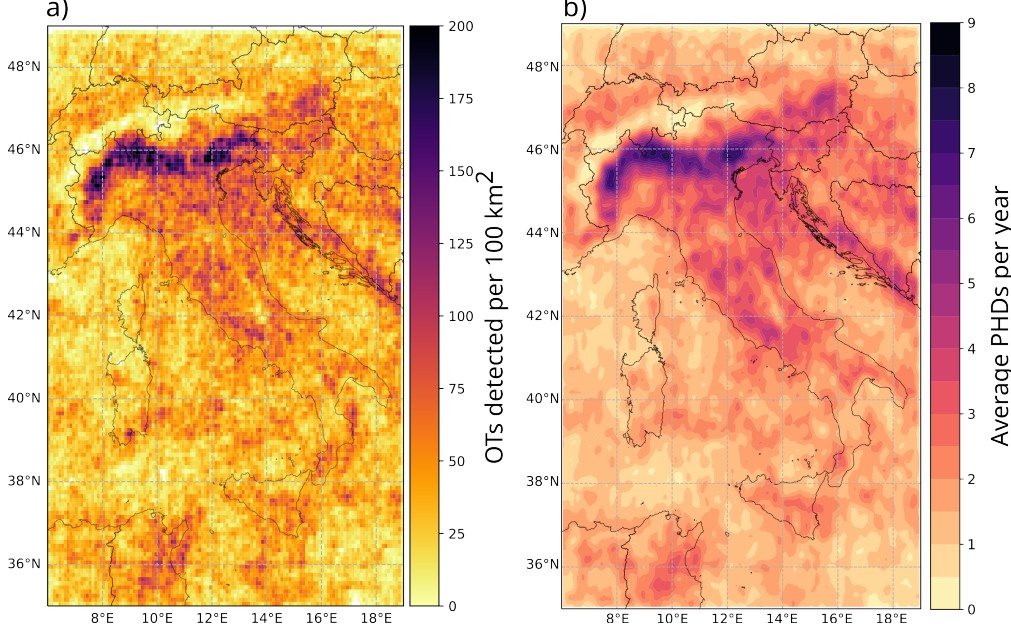

**Figure 6.** a) Same as Fig. 2b, but for OTs retained after the hail-specific filter. b) The resulting average number of potential hail days (PHDs) per year over 2016-2020 estimated from the hail-related OTs distribution in a) after spatial smoothing with a Gaussian filter.

The intra-annual variations in hail frequency are estimated on a monthly basis in terms of the geographical distribution of PHDs per month (Fig. 7), and with histograms of hail-related OTs over the whole domain separately for land and sea areas (Fig. 8). Hail frequency is found to be almost zero in early spring, but increasing from April to May, when the cooler temperatures over land and sea surface lead to lower low-level moisture, which limits the development of deep moist convection. Hail likelihood rapidly increases in June and July over continental areas, with a well-defined peak around the Alpine region. Particularly in July, besides the widespread maximum over the southern pre-Alps in northern Italy, circumscribed hotspots over central Switzerland and south-western Germany are detected, in accordance with Nisi et al. (2016). Starting from August and extending to September, a significant reduction in hail-filtered OT rate over land is evident (Fig. 7e-f and 8a), coupled with a gradual increase in thunderstorm development over the warm waters of the Tyrrhenian and Adriatic seas (Fig. 8b). Finally, in October (Fig. 7g) a further shift of hailstorm activity towards lower latitudes of the southern Mediterranean sea is detected, while maintaining the hail hotspot over the Thyrrenian sea. This is linked to the increased cooling of the continental surface and the growing likelihood of mid-latitude cyclone formation in this region resulting from the maintenance of warm sea surface temperatures (e.g., Flaounas et al., 2022).

Hail is found to be generally more frequent over land during daytime (from 8 to 19:45 UTC, Fig. 9a) and over the sea during nighttime (from 20 to 7:45 UTC, Fig. 9b). Hail likelihood during daytime is highest over southern pre-Alpine areas and significantly pronounced over high-elevation terrains, especially in the eastern continental part of the domain (Austria, Slovenia, and the Balkans) and over the central-southern Italian peninsula. During nighttime, the north Adriatic sea is an



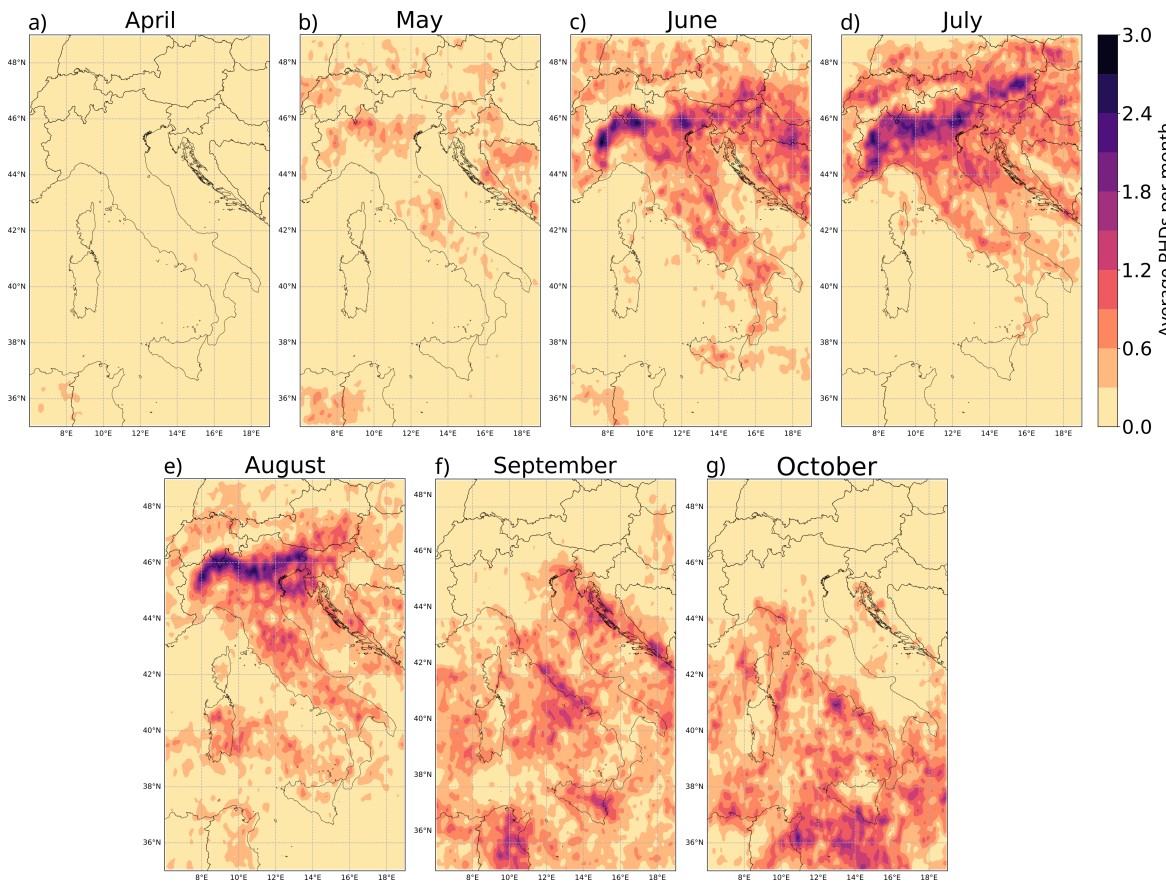

**Figure 7.** Same as Fig. 6b, but separating among a) April, b) May, c) June, d) July, e) August, f) September, and g) October.

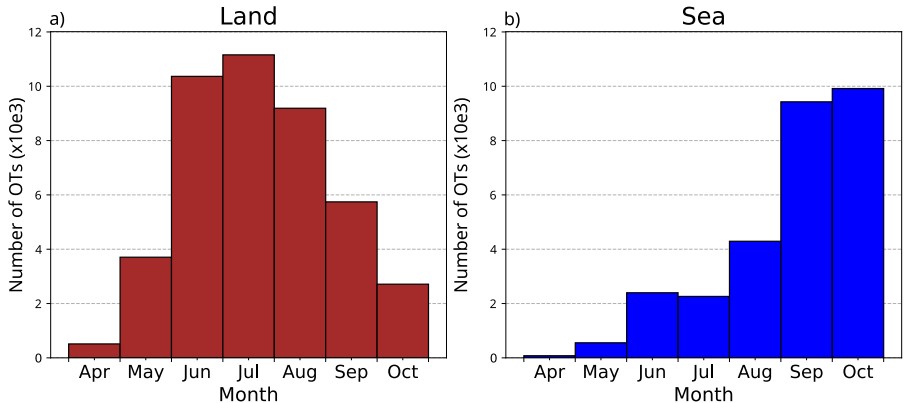

**Figure 8.** Distributions of hail-related OTs per month separately over land (panel a) and sea (panel b).

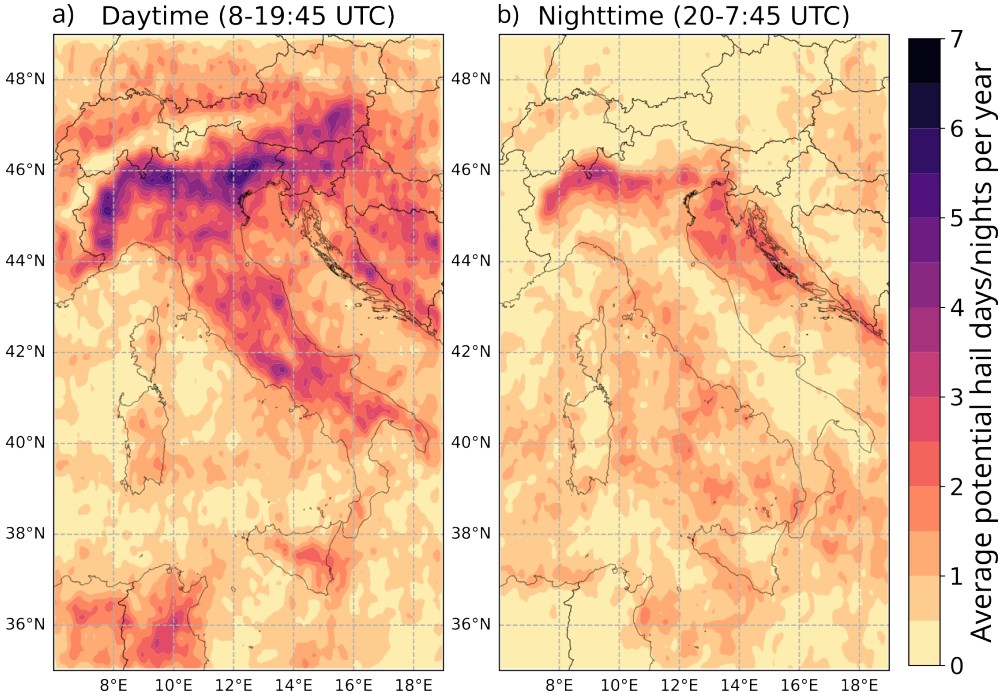

**Figure 9.** Same as Fig. 6b, but separating between a) Daytime (i.e., 8-19:45 UTC) and b) Nighttime (i.e., 20-7:45 UTC).

evident hail hotspot, with the maximum along the western Croatian coast. This may be linked to the combination of the north-eastern mountains, supporting convective development, with local near-surface wind convergence, causing the formation and organization of convective cells over coastal areas during the afternoon and early evening (Mikuš et al., 2012; Jelić et al., 2020).

A further relevant nighttime hotspot is detected over north-western Italy along the border with Switzerland and all along the pre-Alpine southern flank area. This is presumably linked to late-evening thunderstorm formation over the foothills, most likely imputable to katabatic winds interacting with thermally-driven Alpine Pumping circulation (Bica et al., 2007). The interaction between these flows produces local convergences, enhances vertical wind shear and orographic lifting, and ultimately promotes convection initiation over the region (Nisi et al., 2020). Finally, the western Italian coast also shows prominent (to a lesser

extent) nocturnal potential hail signals, which are mostly underestimated by ESWD-based estimates (Fig. 1d) likely owing to the reduced observational activity during nighttime.

Figure 10 shows the diurnal cycle of hail-related OT activity separated between land and sea areas. Over land, very infrequent OT detections are revealed during the night and early morning, with a rapid increase starting from 10 UTC and peaking at 15 UTC (i.e., 17 CEST). This reflects the maximum diurnal heating of the near-surface troposphere, which reduces convective

inhibition and increases the likelihood for atmospheric instability conditions. During the afternoon, a slightly more gradual decrease is detected. Over the sea, OTs are more likely to form during the night and early morning (from 23 to 9 UTC) compared to land, with a local maximum at 3 UTC. This is most likely linked to the north-eastern Adriatic hotspot of nocturnal





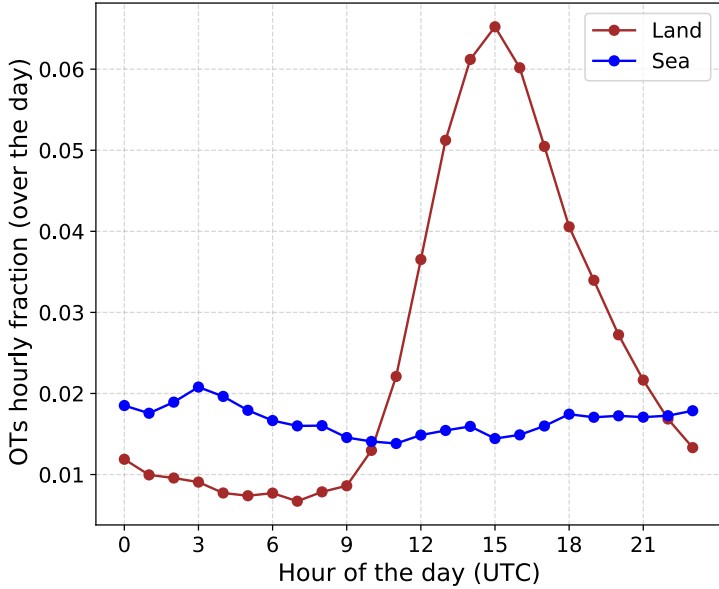

**Figure 10.** Hourly fraction of hail-related OTs separating land (in red) from sea surface (in blue) and aggregating over the whole spatial domain.

hailstorm generation detected in Fig. 9b. Afterwards, a gradual decrease in marine OT activity is detected after around 12 UTC, when OT frequency slightly fluctuates during the afternoon and increases during the evening.

These findings are in good agreement with the spatio-temporal distribution of the European OT characterization (over 2004-2009) of Bedka (2011).

## 4.2 Matching with ESWD hail reports

The appropriateness of the hail proxy is evaluated by comparing both filtered and unfiltered OT detections with ESWD hail reports. The aim is to quantify how often an OT is found in the spatio-temporal proximity of observed and confirmed hailstorms.

Despite the same ESWD sample has been used to define the environmental thresholds of the reanalysis parameters used to filter the OTs (Sect. 3), the comparison between reports and hail-related OT detections is believed not to be affected by any substantial overfitting issue given the full independence of the two datasets. Unfortunately, a complete assessment of the performance is not possible due to the partial description of the real hail occurrence available with crowd-sourced reports, as discussed in Sect. 2.1. While confirmed reports can give precious spatio-temporal information in case of actually occurred hailstorms, their lack

does not necessarily imply the non-occurrence of events. Several events may be missed or under-reported in the crowd-sourced data (e.g., in sparsely populated areas such as mountainous regions, or during nighttime), which is recognized to be a dominant issue of the ESWD database (Púčik et al., 2019). Statistically, this implies that in the contingency table we can only assess the hit and the miss rates describing the joint distribution of the "forecast" (OT detections) and observations (hail reports), and not the false alarms or the correct negatives.



**Table 2.** Comparison between OT detections and ESWD reports (with quality level QC1 or superior) considering the two spatio-temporal matching of 25 km / ±1 h and 75 km / ±3 h, both for the original "Orig OT" and the hail-filtered "Filt OT" datasets, only over land. The fractions of ESWD reports matching OT detections (Hit ESWD rep. row), and the fractions of OTs hitting at least one ESWD report (OTs hitting ESWD row) are reported.

| Spatio-temporal matching | 25 km / ±1 h | | 75 km / ±3 h | |
|---|---|---|---|---|
| | Filt OT | Orig OT | Filt OT | Orig OT |
| Number of ESWD rep. | 2,293 | | 2,293 | |
| Hit ESWD rep. | 1,410 | 1,552 | 1,788 | 1,934 |
| | 61.5% | 67.7% | 78.0% | 84.3% |
| Number of OTs | 433,862 | 597,547 | 433,862 | 597,547 |
| OTs hitting ESWD rep. | 12,501 | 14,158 | 45,868 | 52,643 |
| | 2.88% | 2.37% | 10.57% | 8.81% |

The matching condition between satellite detections and hail reports consists of a temporal window of ± 1 h around the OT detection time and 25 km from each OT location. This relatively sharp temporal window is considered given the high temporal accuracy ≤ 1 h characterizing more than 98% of ESWD reports (Fig. 1b). The 25-km distance criterion is retained from Bedka (2011) and Punge et al. (2017) and accounts for a maximum storm motion of 60 km h$^{-1}$ and possible latitude/longitude uncertainty for ESWD reports. The matching with the set of hail reports having a quality level QC1 or superior (2,293 reports) is
considered for both the original and the hail-filtered OT datasets (over land only), to highlight possible differences owing to the filter procedure. Additionally, to investigate the sensitivity of the matching conditions between the hail proxy and observations, a less conservative spatio-temporal constraint of ± 3 h over 75 km is proposed. The results are reported in Table 2.

With respect to the 25 km / ±1 h matching, an OT of the unfiltered dataset is found in the vicinity of 67.7% of the ESWD hail reports, while for hail-filtered OTs, the hit rate reduces to 61.5%. To give an objective evaluation of these results, a comparison
with the similar analysis previously performed by Punge et al. (2017) is proposed. However, given several differences in the research design, it is necessary to focus only on the qualitative aspects. The main differences between the present study and that of Punge et al. (2017) are: a larger spatial domain covering all of Europe (extending from England to Russia and from Norway to Egypt) over ten years (2004-2014), the consequent different spatial coverage in the distribution of ESWD reports (which are only scarcely available in some regions such as France or Portugal during their analysis period), the employment of
the former non-probabilistic version of the OT detection algorithm from satellite infrared imagery, and the substantially coarser description of hail-favoring convective environments owing to ERA-Interim reanalysis. Using a spatio-temporal window of 25 km and ± 1 h around each OT detection, they found a hit rate decreasing from 40.3% to 39.7% for the unfiltered and filtered OT datasets, respectively, for a subset of 2,475 ESWD reports with quality level QC1 or superior and temporal accuracy ⩽ 15 min. This indicates that the proxy obtained in the present analysis has improved by roughly 22% over that obtained by Punge et al.
(2017) in terms of match with hail reports. The main reason for this improvement is likely the new probabilistic OT detection algorithm of Khlopenkov et al. (2021) allowing the identification of weaker looking satellite features that were previously harder to detect with fixed temperature thresholds. Further, this is also accompanied by a reduction of false OT detections. As a



possible indication of this, the fraction of OTs found in the vicinity of hail reports by Punge et al. (2017) increased from 0.67% to 0.84% after the filter, conversely to the detected values of 2.37% and 2.88%, respectively (Table 2). Considering the more generous spatio-temporal window of 75 km / $\pm 3$ h, a decisive increase for all hit rates is noted: the fractions of hail reports detected by the filtered and unfiltered OTs sets are 78.0% and 84.3%, and the respective OT rates matching ESWD are 10.57% and 8.81%.

Even if an increase in the fraction of OTs associated with hail reports is detected, its absolute value remains low. This may be related to under-reporting issues in the ESWD dataset, or the limits of the conservative design of the filter proposed. The relevance of the under-reporting problems of the hail database is highlighted by the strong year-to-year variability in the fractions of filtered OTs found in the vicinity of ESWD reports (not shown). Considering the 75 km / $\pm 3$ h matching, in the year with the largest number of reports (2019, with 659 reports) the fraction of OTs in their vicinity exceeds the 18%. Conversely, when the lowest number of hail reports were issued (2016, with 298 reports) the amount of matching satellite detections is slightly less than 7%. With respect to the limits of the OT filtering procedure designed, the aim of the minimum conditions imposed is to remove all possible situations that are unsupportive to hail development in a thunderstorm. The retained environmental characteristics and the associated OTs are shared also by storms producing other severe weather than hail. A potential way forward to more sharply discriminate specifically hail environments within this superposition could be to include additional hail-related observations to expand the sample of ambient conditions, such as hailpad records. However, hailpad networks cover only a smaller part of the selected region, which prevents a substantial enlargement of the validated OT data sample. Therefore, ESWD still represents the best available dataset for ground-truth hail occurrence.

## 4.3 Hailstorm environmental signatures

To better understand under which conditions hail reports are correctly identified or missed by the OT-based approach, the associated environmental conditions described with the CP reanalysis predictors are investigated. Parameter distributions are analyzed separately for hit or missed reports. Further, to take into account the role of hail severity, only the subset of 2,249 ESWD reports with information on the maximum hailstone size is considered (in the following referred to as ESWD-S), and results are reported separating between small ($<$ 3 cm) and large hail ($\geqslant$ 3 cm). For some hailstorms more than one report is issued, which share the same ambient conditions at a specific temporal stage of the storm. For this reason, duplicate values of the parameter distributions that at the same hour present exactly the same values of all five SPHERA parameters are discarded. This is necessary to avoid artificial deviations owing to repetitions of the samples in the resulting distributions. Finally, the satellite-measured cloud-top thermal characteristics in presence of ESWD-S reports are analyzed to further detail hailstorm ambient features.

The cumulative density functions of SPHERA parameters in the presence of ESWD-S reports, separated into four categories based on matching and hail severity conditions, are presented in Fig. 11. 66% reports are successfully detected by the filter-based proxy, the majority of which (69%) pertains to large hail, while missed cases are similarly associated with small (46%) and large hail (54%). By increasing the hailstone size, the distributions tend to shift towards values with enhanced potential for severe convection (i.e., larger CAPE, K index, and DLS, and smaller SLI) and higher $H_0$. The most evident separation for all




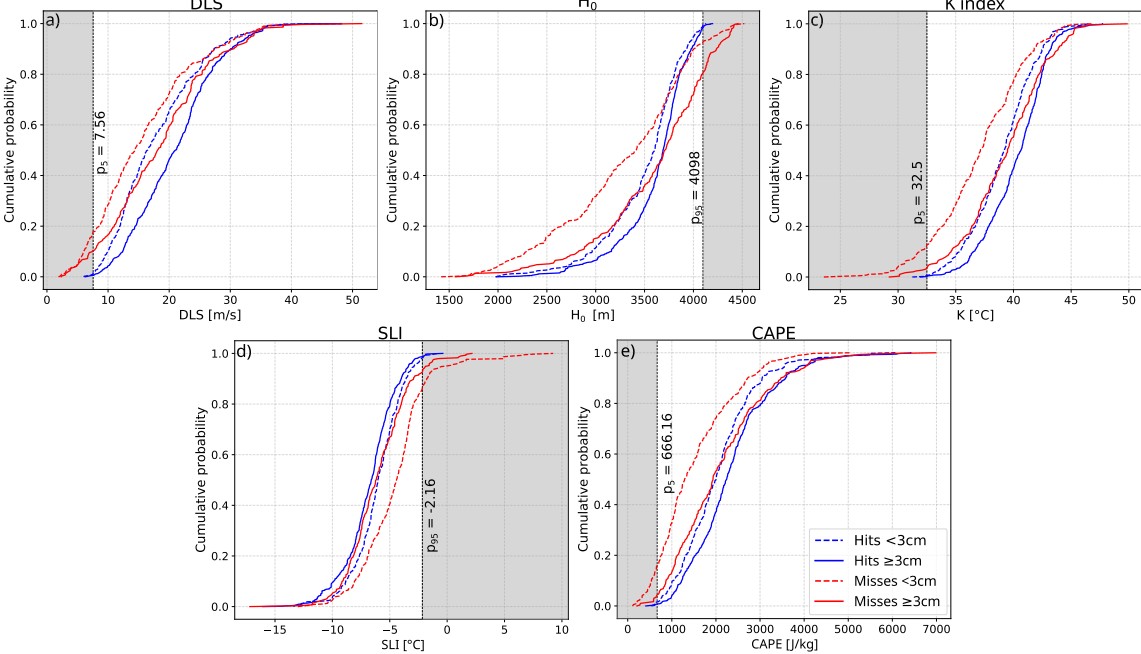

**Figure 11.** Cumulative density functions of the five parameters selected from SPHERA in presence of ESWD-S hail reports. The same criteria described in Sect. 3 apply to spatio-temporally aggregate the parameters in the vicinity of hail reports. Reports are divided into small hail (< 3 cm, dashed lines) and large hail ($\geqslant$ 3 cm, solid lines) when hit (in blue) or missed (in red) by the hail-specific OT dataset. The black dashed vertical lines indicate the thresholds identified for filtering (Table 1). The shadowed portion of the distributions show when the filter is active. a) DLS, b) $H_0$, c) K index, d) SLI, and e) CAPE.

parameters (including DLS, but to a lesser extent) emerges for the missed–small hail class (Fig. 11 - dashed red lines), showing cumulative density curves systematically shifted towards less unstable, less sheared and warmer environments. Interestingly, only 4% of hit ESWD-S reports show at least one parameter falling in its filtered data range (i.e., shadowed areas in Fig. 11),

while the fraction decisively increases to 42% for missed reports.

To investigate the inter-relationships between ambient descriptors, hailstorm severity and matching conditions, the parameters spaces for the four hail reports classes are considered in the form of bi-variate histograms. Figure 12 shows the joint distributions of $H_0$ and K index for the four hailstorm classes. A joint increase of freezing level height with atmospheric instability and low-level moisture content is noted, suggesting a positive linear relationship between $H_0$ and K index. The distributions

for hits (12a-b) are compact and do not present relevant differences among hail sizes. On the other hand, misses counterparts (12c-d) extend over wider ranges and show evident shifts between small and large hail (with $H_0$ and K index medians greater by roughly 300 m and 2.3°C, respectively). This suggests that missed-large hail events are characterized by generally warm vertical atmospheric profiles (with $\sim$ 20% freezing level heights above the imposed threshold), while missed-small hail tends to form in lower-instability and colder ambient conditions.




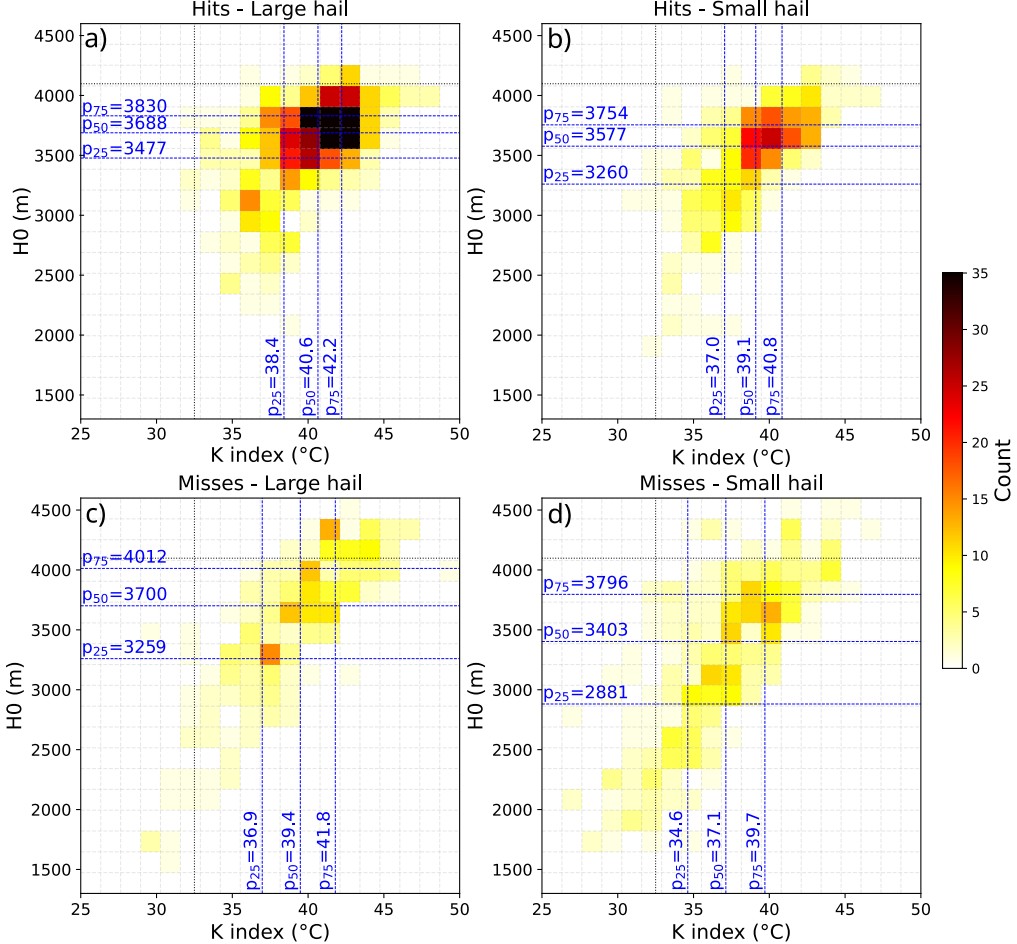

**Figure 12.** Bi-variate histogram distributions of $H_0$ vs K index in presence of ESWD-S hail reports for the separation considered in Fig. 11: a) hits–large hail, b) hits–small hail, c) misses–large hail, and d) misses–small hail. The blue dashed vertical and horizontal lines represent the median ($p_{50}$) and the interquartile (IQR) range values ($p_{25}$ and $p_{75}$) of the distributions. The black dotted lines report the thresholds used for the filter defined in Table 1.

More dispersion characterizes the joint $H_0$-DLS distributions (Fig. 13). In all four classes, DLS covers a broad spectrum with interquartile ranges (IQRs) of $\sim$10 m s$^{-1}$, confirming the difficulty in separating events by their hailstone sizes through the vertical wind shear (Kunz et al., 2020). The difference of roughly 7 m s$^{-1}$ in median DLS values from misses-small hail (14.39 m s$^{-1}$) to hits-large hail (21.24 m s$^{-1}$) suggests the increase in hail severity with storm organization.

Significant spread characterizes also the CAPE-DLS spaces describing the relationship between atmospheric instability and storm organization (Fig. 14). Also in this case, the most different conditions emerge for the missed–small hail class, characterized by generally pronounced low-CAPE (median 1,286 J kg$^{-1}$) and low-DLS environments.



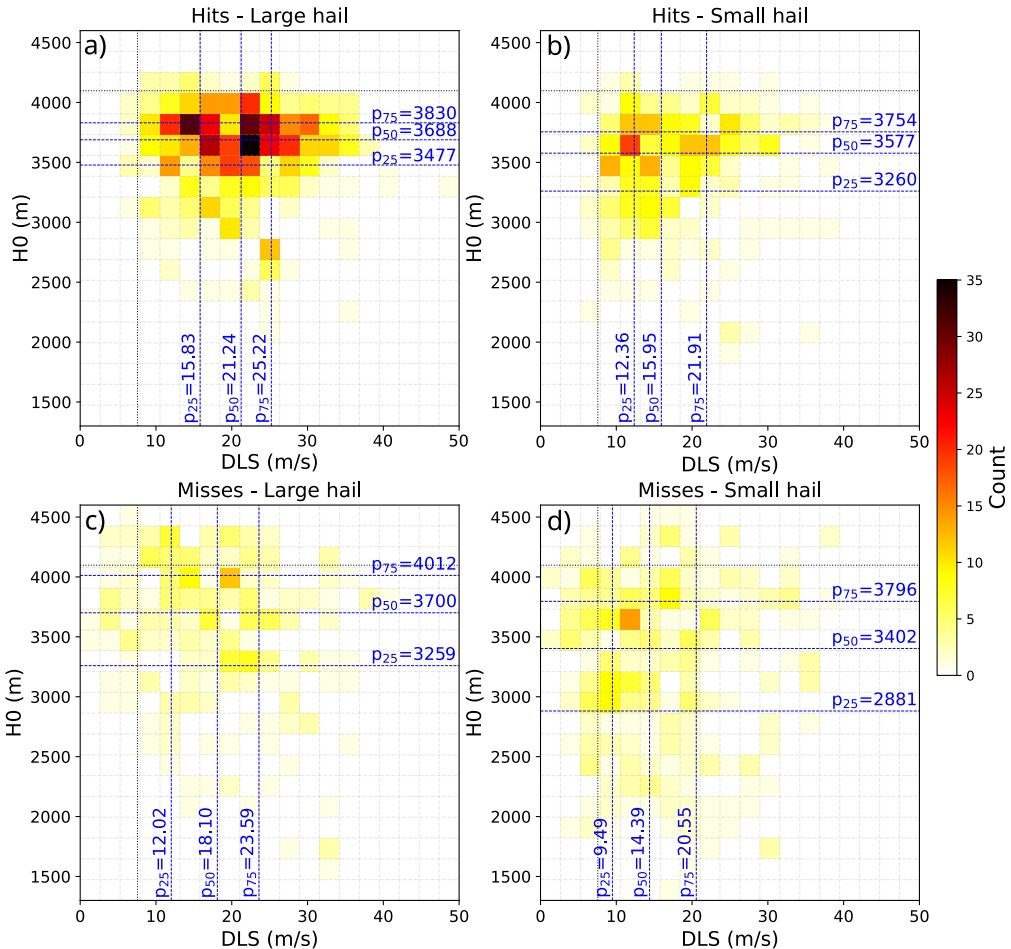

**Figure 13.** Same as Fig. 12, but for $H_0$ and DLS.

A factor playing a central role in the identification of an OT from satellite scans data is the thermal characteristic of the cloud top where the OT can be found. Previous research showed how OTs linked to deep convective clouds can be detected as cold pixels in infrared satellite imagery scans (e.g., Morel and Senesi, 2002; Mikuš and Mahović, 2013). These cold spots are

associated with small and sharp infrared brightness temperature (IRBT) minima that are near to or colder than the tropopause temperature associated with the anvil cirrus cloud. Hence, a critical variable included in the Khlopenkov et al. (2021) algorithm for automatic OT detection is the temperature difference $\Delta T$ between infrared brightness and tropopause temperatures. A large $\Delta T$ (> 6 K) indicates a penetration of the updraft through the anvil of at least 1-2 km (Griffin et al., 2016). The investigation of the cloud-top thermal conditions in the presence of actually occurred hailstorms could help understand why these have been

correctly identified or not with the OT filter approach. For this reason, the minimum IRBT and $\Delta T$ distributions in the presence of ESWD-S reports are considered (for any OT probability of occurrence and not only for >50% as imposed up to now). The distributions are separated among hit and missed reports for small, large, and very large hail occurrences (Fig. 15). Sharp



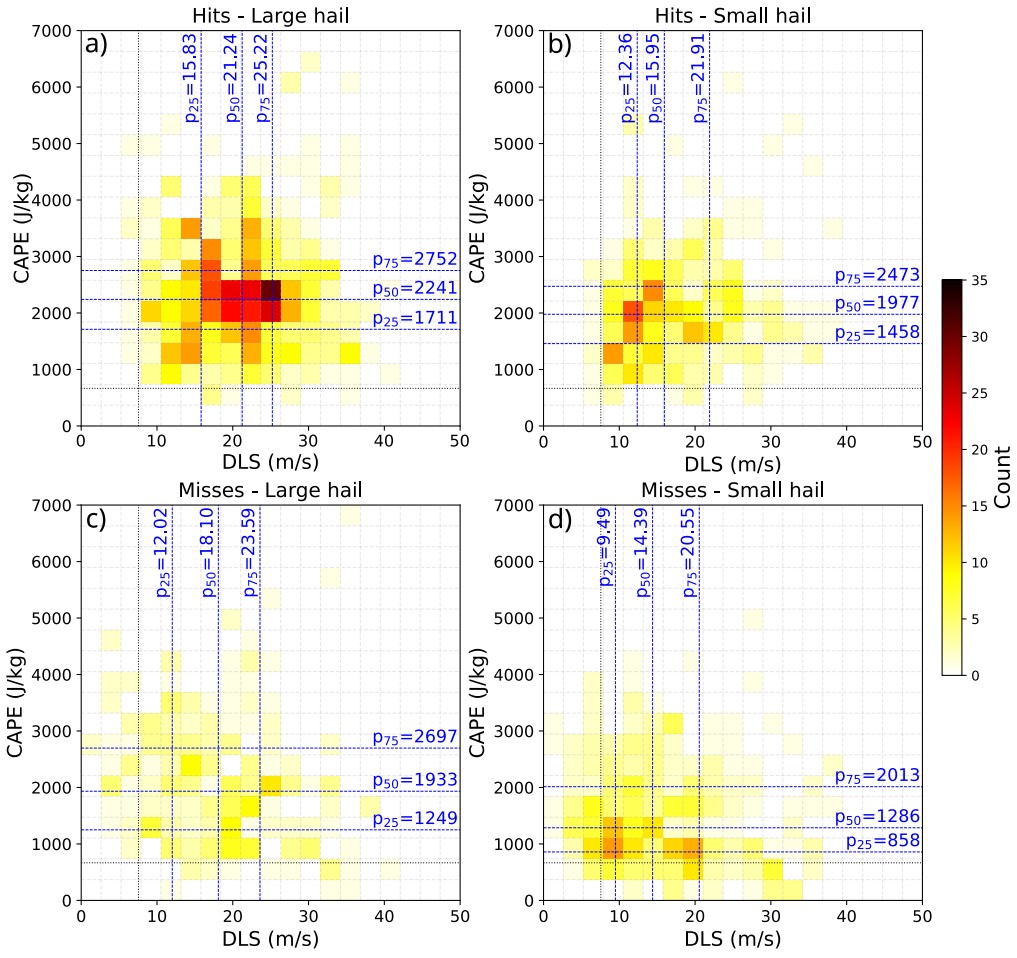

**Figure 14.** Same as Fig. 12, but for CAPE and DLS.

IRBT minima distributions characterize hit reports of all hailstone size (Fig. 15a), with mean values of ∼211 K and rarely exceeding higher temperatures than 224 K. The relative $\Delta T$ minima (Fig. 15c) show almost no positive values, meaning that

IRBT is almost always warmer than the tropopause temperature. Further, the central values of all $\Delta T$ populations are below -4 K, as expected from severe thunderstorms producing prominent OTs. Missed reports (Fig. 15b) present more blunted and higher IRBT minima distributions, extending to temperatures as high as 239 K. The associated mean values suggest a more pronounced separation among hail severity classes, especially in case of very large hail (∼5 K colder than for small hail). $\Delta T$ minima (Fig. 15d) confirms and strengthens these results: the majority (i.e., 54%) of missed ESWD reports are associated with

positive $\Delta T$, reaching values as large as +15 K. These conditions indicate tropopause temperatures substantially lower than those of the detected OT, suggesting not prominent IR signatures that can be expected to be detected. The enhanced separation in $\Delta T$ distributions between small and very large hail, the latter being on average more than 3 K colder, indicates the difficulty for large hailstones to form in these environments.





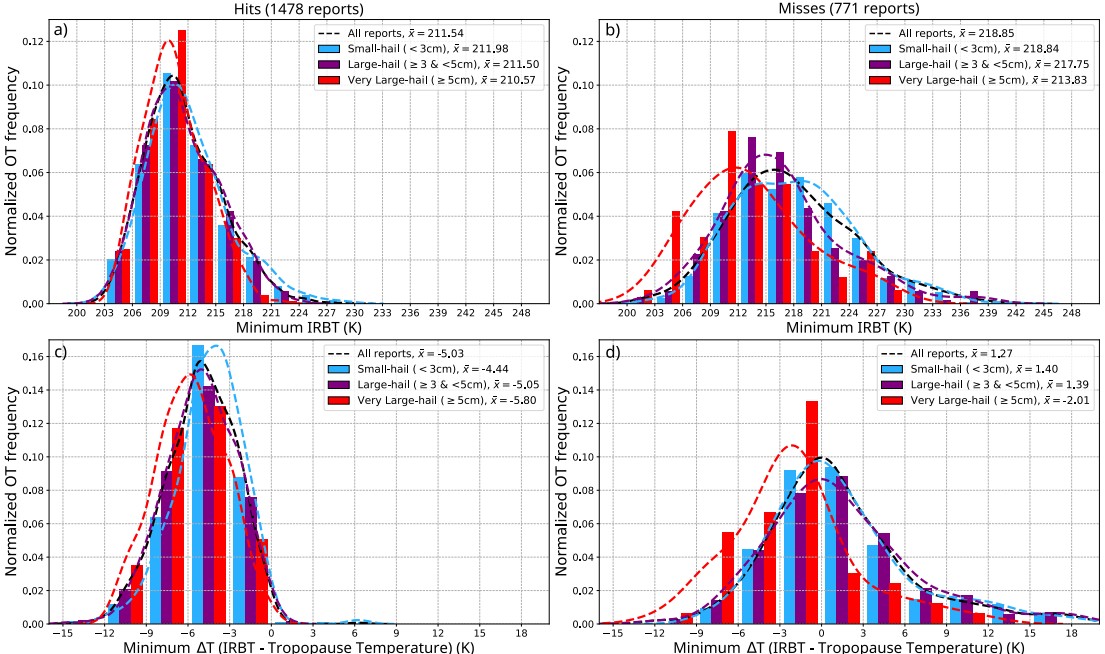

**Figure 15.** Normalized distributions of minimum IRBT (a b) and $\Delta T$ (c d) in presence of hit (a c) and missed (b d) hail reports. The histogram bars quantify the normalized frequency of OTs in the presence of the ESWD-S subset for small (light blue), large (purple), and very large hail (red). Indicated are the mean values ($\bar{x}$) for each distribution. The kernel-density estimated probability density functions are shown with dashed curves in respective colors, additionally, the density functions for the whole ESWD set (including also those reports without maximum hailstone size information) are displayed with black dashed curves.

## 5 Discussion and conclusion

A method for hailstorm identification obtained by combining convection-permitting SPHERA reanalysis environmental predictors, satellite MSG OT detections, and crowdsourced ESWD hail reports has been presented. The analysis over 2016-2020 during the extended summer season (April-October) allows to assess the appropriateness of the hail proxy over south-central Europe, and to investigate the environmental conditions associated with hail. The proxy is based on a filter to identify convective updrafts potentially linked to the formation of hailstones in a thunderstorm by considering the surrounding environment.

Five numerical predictors, quantifying key ingredients for hail development (i.e., most unstable CAPE, K index, SLI, DLS, and freezing level height), are employed to separate OT detections. Single predictors give different spatio-temporal contributions in the identification of hail-related conditions, and their joint use enables to single out satellite-detected updrafts where hail is possible. Indeed, the resulting hail proxy shows a maximum hail potential over northern Italy pre-Alpine areas in June and July peaking at 15 UTC. A hail-related OT is found in the vicinity of 61.5% of ESWD reports, exceeding roughly 22% more

than the previous OT-filter estimate over Europe (Punge et al., 2017), and suggesting an improved appropriateness of the new method. Enhanced suitability of the proxy is observed in case of severe hailstorms: the majority (69%) of correctly identified



reports are linked to hailstones exceeding 3 cm diameters. Furthermore, the analysis of the ambient conditions for different hail severity classes suggests the tendency for large (small) hail to form in environments with higher (lower) instability and wind shear, and within warmer (colder) atmospheric vertical profiles, as expected. However, considerable spread in the associated parameters distributions is found, especially for the missed-small hail class, which also shows the most distinct environmental signature. Additionally, the analysis of the observed thunderstorm top temperature minima reveals systematically warmer conditions when ground hail reports are failed to be identified. The greater challenge for the satellite-based algorithm to detect less prominent OTs characterizing these occasions explains, at least partly, why most of the missed ESWD reports went undetected.

Generally good agreement with recent hail climatologies over the study region is found (Punge and Kunz, 2016). The least hail-prone area over the whole spatial domain is the main Alpine crest, owing to the difficulties for organized convective systems to develop in extremely complex terrains, and is in agreement with radar- (Nisi et al., 2018, 2020) and lightning-based (Manzato et al., 2022b) climatologies over the Alps. Pre-mountainous regions over the eastern Alps show enhanced likelihood for hail formation as described by the OT-hail proxy, with local maxima over south-eastern Austria and Slovenia, in good accordance with Svabik et al. (2013). Moderate hail frequency is detected in southern Germany, which is considered a main European hotspot for hail hazard (Punge et al., 2014, 2017; Fluck et al., 2021); this may be caused by the limited temporal extent of the analysis. The northern Adriatic sea represents the primary marine hotspot for hailstorms, particularly enhanced along the Croatian coastline during nighttime in late summer (August-September), similar to Jelić et al. (2020). The most favorable conditions for hail are found along the Italian pre-Alps, but the potential for hailstorm formation is met throughout north-central Italy. This agrees with several hail climatologies on the national (Baldi et al., 2014), or regional level over north-western (Davini et al., 2012) and north-eastern Italy (Giaiotti et al., 2003; Sartori, 2012; Manzato et al., 2022a). On the seasonal scale, the detected intra-annual variability well agrees with the recent Italian ERA5-and-ESWD-based hail characterization of Torralba et al. (2023) over 1979-2020. Good temporal matching is also found with the ESWD reports statistics during 1990-2018 over Europe (Púčik et al., 2019).

The hail-related environmental conditions identified and the relative tendencies depending on hailstone dimensions are also in line with previous findings. Torralba et al. (2023) found CAPE > 900 J kg$^{-1}$ and a median K index of 30°C for hail occurrence, with increasing values with hail severity, indicating enhanced instability for severe hail development, as expected (e.g., Púčik et al., 2015; Marcos et al., 2021). Kunz et al. (2020), separating between <3 cm and ⩾5 cm hailstones, identified scattered DLS distributions ranging ∼0-30 m s$^{-1}$ for the former and ∼5-30 m s$^{-1}$ for the latter, confirming a high degree of dispersion between hail severity and storm organization, and corroborated by proximity radiosounding data (Púčik et al., 2015). The CAPE-DLS space has been used in numerous studies as a proxy for hail. Wide ranges in the joint distribution have been found by Púčik et al. (2023) and Púčik et al. (2015), but with a clear lack of severe events in low-CAPE and low-DLS environments (which tend to concentrate on the opposite high-CAPE and high-DLS range), similarly to the tendency found in this analysis, as well as by Taszarek et al. (2020). The freezing level characterization is also in accordance with previous studies: Jelić et al. (2020) found an upper limit of ∼4000 m above which no hail on the ground has been observed. Finally, the positive relationship between freezing level height and instability, for which large hail can form in stabler environments if H$_0$



is low while more instability is needed if $H_0$ is higher, has been detected also by Prein and Holland (2018).

The proposed method has demonstrated enhanced appropriateness for the identification of large hail-producing hailstorms.
This is revealed by the large fraction of hit ESWD reports having large maximum hail diameters and sharper distributions of their environmental characteristics. In case of smaller hailstones, a larger degree of uncertainty is revealed by less explicit OT signatures related to higher cloud top temperatures and by environments with lower freezing levels and instability (associated with weaker updrafts that are less evident from the OT perspective). Hence, the fixed thresholds of the high-resolution reanalysis parameters introduced to filter the OTs could mask these occasions. Anyhow, since the primary interest is to enhance the
characterization of the most damaging hailstorms, the developed methodology has demonstrated to be appropriate for this scope.

The five years considered constitutes one of the main limitation of the study owing to the large year-to-year variability associated with hail. Indeed, a period of few years is insufficient for a robust assessment of hail frequency. However, it is appropriate for the purpose to present the novel methodology and to assess its potential. Possible future temporal extensions of
the analysis could decisively enhance its robustness with the purpose to develop a sound climatology for hail.

The imperfect OT observations with the MSG IR satellite instrument constitutes a further source for improvement. Indeed, the majority of missed ESWD reports are associated with detected cloud top temperatures that are higher than those at the tropopause, which are more challenging to be identified with the automatic algorithm of Khlopenkov et al. (2021) applied to the MSG SEVIRI. This difficulty, as noted by Cooney et al. (2021), may be linked to satellite scans performed when storm
tops are not optically thick enough at the initial stages of the storm development, resulting in outgoing radiation being scanned in warmer and deeper regions of the cloud than the updraft top, likely owing to the insufficient density of the particles in this region (Sherwood et al., 2004). This results in smaller temperature differences between tropopause and cloud top, ultimately reducing the probability of OT detection and, consequently, causing to miss the hail event. Hence, possible ways to overcome this issue and to improve the OT detection methodology are the inclusion of visible channel textures together with IR detections
(Bedka and Khlopenkov, 2016), or the enhancement in the spatio-temporal resolution of the satellite scanning that in this study are limited to 3 km and 15 minutes, respectively. Particularly, a higher temporal frequency for detecting thunderstorm OTs, such as that possible with the Geostationary Operational Environmental Satellites (GOES) 16 or 17, or with the novel Meteosat Third Generation (MTG) satellite, is essential, given the rapidity of their formation and dissipation, which may be even below 15 minutes (Elliott et al., 2012).
In this study high-resolution numerical simulations are included to describe atmospheric ambient conditions, which is considered a promising avenue of development in hail research (Allen et al., 2020). Indeed, coarser global datasets have been generally employed in similar studies, whose simulations include physical parameterizations to account for deep moist convection, implying potential significant errors and inaccuracies (Prein et al., 2015). The new regional reanalysis SPHERA is considered in this work, whose horizontal grid spacing of 2.2 km allows to switch off parameterization schemes, as well as a
finer representation of the atmospheric and topographic details. However, a quantification of the added value introduced by this innovation was not feasible, and, generally, no quantitative studies have been performed so far to assess the benefits of finer



spatial grid spacings configurations over coarser datasets specifically for the reproduction of hailstorm environments. Hence, possible future studies could analyze in detail the sensitivity on the driver dataset describing hail environmental predictors to better understand the role of km-scale simulations in this context.

*Code availability.*  The python scripts developed to process and visualize reanalysis parameters, OT detections and hail reports datasets are freely available at https://github.com/agiord/hail-analysis.

*Data availability.*  SPHERA reanalysis data are stored at the ARPAE-SIMC repository and available from the corresponding author upon request (antonio.giordani3@unibo.it). Overshooting top detections data have been made available by NASA (contact K. Bedka for any inquiry: kristopher.m.bedka@nasa.gov). The access to ESWD hail reports was partly granted by ADA Life Project funding under contract
LIFE19 CCA/IT/001257 under the LIFE programme of the European Commission, and partly by the Karlsruhe Institute of Technology (KIT) in collaboration with ESSL.

## Appendix A: Thermodynamic parameters

In the following are reported the formulations of the thermodynamic parameters selected from SPHERA reanalysis:

– **MU CAPE**:

$$CAPE = g \int_{z_i}^{LZB} \frac{T_{v_{parc}} - T_{v_{envir}}}{T_{v_{envir}}} dz \tag{A1}$$

where $g$ is the gravitational acceleration, $z_i$ is the altitude in the lowest 300 hPa where the equivalent potential temperature is at its maximum (i.e., most unstable conditions), $LZB$ is the level of zero buoyancy (or equilibrium level) where the virtual temperature of the parcel $T_{v_{parc}}$ equals the virtual temperature of the surrounding environment $T_{v_{envir}}$. CAPE represents the integrated amount of work over the vertical air column exerted by the upward buoyancy force over the air
parcel.

– **K index**:

$$K = (T_{850hPa} - T_{500hPa}) + Td_{850hPa} - (T_{700hPa} - Td_{700hPa}) \tag{A2}$$

where $T_{xxhPa}$ and $Td_{xxhPa}$ are respectively the temperature and the dew-point temperature at the isobaric level $xx$ hPa. The terms entering the K index represents respectively the lapse rate, the low-level moisture content, and the moist layer
depth.

– **SLI**

$$SLI = T_{500hPa_{envir}} - T_{500hPa_{parc}} \tag{A3}$$



The SLI evaluates the temperature difference between the environment at 500 hPa and a parcel lifted dry adiabatically from the surface to the lifting condensation level and pseudo-adiabatically to 500 hPa.

*Author contributions.* **Antonio Giordani**: data curation, formal analysis, investigation, methodology, software, visualization, writing - original draft. **Michael Kunz**: conceptualization, formal analysis, methodology, resources, supervision, validation, writing - original draft. **Kristopher M. Bedka**: conceptualization, methodology, resources, supervision, writing - original draft. **Heinz Jürgen Punge**: conceptualization, methodology, software, writing - original draft. **Tiziana Paccagnella**: funding acquisition, supervision. **Valentina Pavan**: conceptualization, funding acquisition, supervision. **Ines M. L. Cerenzia**: data curation, resources. **Silvana Di Sabatino**: funding acquisition, project
administration, supervision.

*Competing interests.* The authors have no conflicts of interest to declare.

*Acknowledgements.* The work of some of the authors was supported by the projects OPERANDUM (OPEn-air laboRAtories for Nature baseD solUtions to Manage hydro-meteo risks), which is funded by the European Union's Horizon 2020 research and innovation programme under the Grant Agreement No. 776848, ECOSISTER (ECOsystem for SustaInable Transition of Emilia-Romagna) funded by the European
Union's Next Generation EU program - PNRR (Piano Nazionale Ripresa Resilienza), and ADA (ADaptation in Agricolture) under contract LIFE19 CCA/IT/001257 under the LIFE programme of the European Commission. Kunz's participation in this work has been supported by the Helmholtz Association (Research Program "Changing Earth – Sustaining our Future"). Bedka's participation in this project has been supported by the NASA Applied Sciences Disasters Program project award 18-DISASTER18-0008. The first author is thankful to the colleagues of ARPAE-SIMC for the fruitful discussions, in particular to Davide Cesari for the substantial help in pre-processing SPHERA
reanalysis data.



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
