# Peer review of "Characterizing hail-prone environments using convection-permitting reanalysis and overshooting top detections over south-central Europe"

_EGUsphere, 2023_

## Author Comment (AC1)

**Authors replies to reviewer comments RC1**

**Review for EGUSPHERE-2023-2639**

*Characterizing hail-prone environments using convection-permitting reanalysis and overshooting top detections over south-central Europe*

*Antonio Giordani, Michael Kunz, Kristopher M. Bedka, Heinz Jürgen Punge, Tiziana Paccagnella, Valentina Pavan, Ines M. L. Cerenzia, and Silvana Di Sabatino*

We want to thank the reviewer for his/her assessment of our manuscript. In the following we give our answers to the comments and recommendations that have been raised. Reviewer comments RC are **bold**, our replies AR are in *italic*.

**General Comments:**

**RC: This study aims to develop a proxy for hail frequency, as determined by ESWD reports, based on high-resolution regional reanalysis atmospheric properties coincident with satellite OT detections. The authors demonstrate the proxy's improved capability in matching with hail reports relative to previous similar studies performed with coarser global reanalysis, although, by their admission, determined with focus on qualitative aspects. The thorough explanation of SPHERA variables and description of robust quality analysis are appreciated.**

**The concept of the filter could be introduced earlier or more gradually – it felt as though we were suddenly discussing the filter as a given before it was properly explained. That said, it is not convincing that the goal of obtaining a new proxy for hail has been satisfied quantitively. That is, the filter approach may be overly conservative compared to a random forest or other machine learning approach. It is perhaps good at removing cases where hail is unlikely, but not capable of skillful positive predictions. This task of separating events by their hailstone sizes through environmental analysis seems better suited for perhaps a logistic regression or a Bayesian analysis approach – as the layering of independently determined statistical filters seems to be too broad of a solution. Furthermore, adding additional years seems necessary given the high variability of year-to-year hail reporting, as a means to alleviate overfitting concerns, and to help the environmental stratifications.**

**I recommend major revision with consideration of the specific comments listed below. Furthermore, the paper can benefit from grammatical revision – some examples of which are listed as technical corrections.**

*AR: We thank the reviewer for the raised points. We agree that the concept of the filter in the original formulation of the manuscript seemed (at least partly) assumed as known to the reader, hence in the revised manuscript we clearly defined what is considered for "filter" since the Introduction. Additionally, we specified more clearly and unambiguously the aim of the proposed analysis, which is mainly the description of a new methodological approach tailored to exclude non-hail events, while is not foreseen for positive hail predictions nor for the definition of a climatology or long-term statistics for potential hail events, but can potentially be used for the purpose if adequately extended over multi-decadal time periods. In the revised manuscript we highlighted more clearly the objectives at the end of the Introduction section.*
*With respect to the conservative nature of the filter approach developed, we would like to point out*

*that the machine-learning alternatives mentioned by the reviewer might be more appealing in detecting true positive hail occasions in some situations and would certainly be interesting and valuable in the investigation of hail events as demonstrated by recent published literature in this context (e.g., Gagne et al., 2017; Gensini et al., 2021; Torralba et al., 2022; Scarino et al., 2023). A potential proposal for assessing the value of such an approach in this context could be to train a neural network with a large set of different environmental parameters (similarly e.g. to Prein & Holland, 2018), which however we believe is beyond the scope of the present paper, besides it would require an intensive effort. Furthermore, the necessary division into training and testing subsamples that a machine learning analysis would require is not possible for our case due to the short temporal series considered of only five years. Indeed, we are aware of the short time period of the analysis for which including additional years could be beneficial as pointed out by the reviewer. However, the extension of the temporal domain into the future (2021, 2022, 2023,..) is not feasible owing to SPHERA reanalysis temporal constraints (which ranges up to 2020 and for the moment future extensions of the dataset are not envisaged). On the other hand, extending into the past (2015, 2014, 2013...) we believe would not add significant benefits for the aim of the paper owing to the lower number of ESWD reports in the past years, as well as the systematical missing information on maximum hail size for the majority of reports (check e.g. Figure 1b in Torralba et al., 2023, who also show that, over Italy, only in 2019 and 2020 the number of hail reports amounts to more than 50% than those reported in the 20-year period 2000-2020).*

*In the revised manuscript we expanded the possibile future outlook in the Conclusion section, proposing that potential subsequent analyses could include the investigation of hail-related conditions through machine learning approches given the promising results obtained by documented recent literature that has been properly cited.*

*Additionally, we thank the reviewer for flagging many grammatical errors, in the revised manuscript we tried our best to improve the grammatical correctness of the manuscript, and we hope that it gained the desired correctness and clarity.*

**Specific Comments:**

**RC: Lines 101-108: The method for coupling satellite, SPHERA, and hail reports should be previewed somewhere around here in the introduction. "optimally tuning the combination" is mentioned, but in what kind of approach? Furthermore, "optimally tuning" is stated, but that seems to extend as far as layering 5 statistical thresholds, which does not suggest *tuning*.**

*AR: We agree that the term "optimally tuning" could probably be misleading and not completely correct in this context, hence in the revised manuscript we removed the expression and reformulated the highlighted sentence as suggested. Further, we introduced the coupling method and the filter concept more gradually.*

**RC: Fig 1/Line 137/Line 371: 1 h is said to be the cutoff for "high temporal accuracy," but what is the justification for this designation, as it seems like a rather large uncertainty, especially compared to the significantly greater sampling at 15-min temporal accuracy? What is the expectation for how 15-min vs. 1-h temporal accuracy would impact the hail environment characterizations?**

*AR: 1h was considered the threshold for ESWD temporal accuracy to be congruent with the spatio-temporal window (± 1h, 25 km) considered in the spatio-temporal matching criterion with OTs*

*presented in Section 4.2. It must be remembered that overshooting cloud tops observed near the tropopause (indicating strong updrafts and potentially associated with the beginning of hail growth) and hail reports observed at the surface are necessarily characterized by a certain spatio-temporal shift which implies a certain relaxation when matching the occurrence of one phenomenon with the other. However, we agree with the reviewer that for hail observation purposes 1h constitute a large temporal uncertainty, and the choice to consider the ESWD subsample with temporal accuracy ≤15 minutes, which matches also the sampling frequency of the satellite instrument detecting the OTs, could be more appropriate. In any case, it must be pointed out that the largest fraction of reports including information on their temporal accuracy is populated by reports with temporal accuracy ≤15 minutes (2077 reports: 77%), while 575 reports (21%) have temporal accuracy = 30min (421: 15%) or = 1 hour (154: 6%). For consistency, in the revised manuscript we adjusted the highlighted sentence accordingly and considered the ESWD subsample having temp. acc. ≤15 minutes when referring to "high temporal accuracy". Further, we considered both the ESWD subsamples having temporal accuracy ≤1h and those having temporal accuracy ≤15 min separately in the discussion of the statistical evaluation between OTs and hail reports matching.*

**RC: Line 139: "A substantial increase is noted for 2019 and, to a slightly lesser extent, for 2020." – Is this natural, or owed to ESWD hail reporting reasons? Showing more years may reveal more about the expected year-to-year variability of reports.**

*AR: The large increase in hail reporting in 2019 is believed to not be related to natural reasons but to a substantial increase in the public awareness and cooperation among European institutes collecting reports as observed in Púčik et al., 2019 (Section 3.a), and more recently by the ESWD reports statistics published by ESSL demonstrating the exponential increase in the number of hail reports in more recent years (2021, 2022, 2023):* [https://www.essl.org/cms/hailstorms-of-2023/](https://www.essl.org/cms/hailstorms-of-2023/) *(Figure reported below). Further, this statistics offers also a hint of the low impact that extending the temporal period of the analysis backwards could have (as discussed in the reply to the general comment). On the other hand, the difference between 2019 and 2020 is most likely imputable to natural year-to-year variability in hail occurrence over the region. Indeed, 2020 registered the lowest number of OTs (see Table below), which suggests less thunderstorm activity and hence that hail-prone conditions were more unlikely in the study region. In the revised manuscript we included an additional statement to clarify this point.*

[Figure]

| Year | 2016 | 2017 | 2018 | 2019 | 2020 |
|---|---|---|---|---|---|
| N of OTs | 171.947 | 189.613 | 271.581 | 208.859 | 149.042 |

**RC: Lines 193, 200, 237, 238: "… is included in the filter" – Again, reference to "the filter" and "full filter" is presented as a given without establishing what exactly is meant by the word. Reading further into the manuscript helps fill in and confirm the reader's intuition, but it would be helpful if there was a more explicit introduction to the filter concept – perhaps as a paragraph at the start of Section 3.**

*AR: The text has been revised accordingly (see the reply to the general comment and to the first specific comment).*

**RC: Line 257: Please explain what is meant by "the limitation imposed by choice of the $H_0$ threshold."**

*AR: In the revised manuscript the specified sentence has been reformulated to avoid ambiguity and enhance clarity.*

**RC: Lines 264-266: It isn't very convincing that the regions highlighted as having lowest OT removal are distinct from "noise". That is, the Gulf of Taranto and areas southwest of Sicily (among others) look comparably shaded to the ones the authors mention. Perhaps a discretized color bar in increments of 20% would help, but otherwise these discussion points should be removed or reframed in a way that's better suited to the image.**

*AR: We thank the reviewer for the raised point and we appreciate the suggestion. Figure 4 has been re-plotted accordingly (with discretized colobrar, reported below) and substituted in the text. Anyhow we agree with the reviewer: even with 5 discretized levels it is not evident that areas with lowest OT removal mentioned in the text (Po valley, Adriatic and Thyrrenean coastlines) are distinct from "noise" and are similar to other areas mentioned as southwest Sicily or the Gulf of Taranto. Hence, we reformulated the discussion in a more cautious fashion.*

[Figure]

**RC: Lines 271-272: There should be discussion of how the number of OT detections change over these same times of day and months, and how that may influence the interpretation of these fractions.**

*AR: Figure 5 has been expanded by adding two additional plots showing the total number of OTs per hour and month (reported below), and an additional relative discussion has been included in the text.*

[Figure]

**RC: Lines 360-363: It's not seen how "full independence of the two datasets" equates to no concern for overfitting. The authors are relating observed hail reports to coincident environmental analyses, that have spatial dependency, and then evaluating spatially dependent, environmental-analysis-filtered storm signatures. So it isn't convincing that "full independence" applies. The claim should be tested by doing the method but with one evaluation year held out from consideration for the filter. Perhaps even doing that for each year and evaluating the average performance. Adding additional years would also help in addressing this concern.**

*AR: Given the conservative nature of the proxy developed, which mainly focuses on filtering out the occasions where hail is unlikely, we believe that potential overfitting issues related to the usage of the same sample of ESWD hail reports data to define the environmental thresholds of the reanalysis convective parameters and to verifiy environmentally-filtered independent satellite cloud-top detections are unlikely to be dominant. Additionally, it is believed that operating a cross-validation by re-calculating the environmental thresholds leaving out single years/months from the sample analysis would be a considerably time-consuming task that would not change substantially the results owing to the small influence that slightly modifying the 5th or 95th percentiles (defining the thresholds) of the resulting environmental parameters distribution might have. As a kind of hint, we considered in the following the values of the environmental thresholds calculated by aggregating over the years 2016-2018 and 2016-2020 (considered in the manuscript) separately, which choice hardly have an impact in the resulting OT filtering and hence impact on the final proxy for hail:*

| Parameter | 2016-2018 | 2016-2020 |
|---|---|---|
| CAPE_MU | 618.42 J/kg | 666.16 J/kg |
| LI | -2.08 °C | -2.16 °C |
| Kindex | 32.5 °C | 32.5 °C |
| DLS | 9.25 m/s | 7.56 m/s |
| $H_0$ | 4039 m | 4098 |

*However, we agree with the reviewer that "full independence" might be too hard of a statement, hence we revised the text accordingly being more cautious.*

*With respect to the addition into the analysis of additional years we believe it is out of the scope of the present work (refer to the reply to the general comment above). Further, we would like to point out that we explicitly proposed the present analysis only for testing the new hail proxy method (see the Introduction, line 104) rather than for developing a sound and robust climatology or statistics for potential hail which is not possible for 5-year data (see the Conclusion, lines 527-530).*

**RC: Table 2: Was a similar Table for day vs. night and/or month-to-month performance considered?**

*AR: We thank the reviewer for the interesting suggestion. We agree that reporting the inter-monthly and inter-daily variability of the performance between hail-filtered OTs and ESWD reports could be of interest to the reader. For this reason we included a new Figure (reported below) in the new Appendix B reporting how the hit rates change separating month-by-month and between day and night (when considering the spatio-temporal window for matching of 25 km and ± 1h), and included a brief discussion in Section 4.2.*
*Furthermore, as a consequence of this analysis, we thought of re-defining the hour ranges included in "daytime" and "nighttime": in order to try to isolate more appropriately the peak of convective activity over land detected in Figure 10, the daytime temporal range has been shifted to 10-21:45 UTC, and consequently nighttime to 22-9:45 UTC. Figure 9 has been revised accordingly.*

[Figure]

**RC: Line 416: Please explain how it is decided which report with duplicate environmental conditions is kept.**

*AR: In case of multiple hail reports issued for the same storm and presenting the same environmental conditions within the spatio-temporal neighborhood considered for its ambient characterization, the report presenting the largest maximum hail diameter is kept. We are aware that a more detailed and precise information could benefit the environmental characterization, such as that possible by knowing the entire hail size spectrum for any issued hail report, but as of today is not feasible and the best possible information relies on the knowledge of the maximum hailstone size detected, which justifies the adopted choice. A note has been added in the text to explicitly account for this choice as suggested by the reviewer.*

**RC: Line 461: Please justify why "as expected."**

*AR: An appropriate reference has been added (Scarino et al., 2023).*

**RC: Lines 487-488: How can this conclusion be offered when the assessment used OTs to characterize the ΔT in missed cases? That is, if the OT are there and measured to be systematically warmer for missed cases, how can it be said that prominent OTs were not detected to characterize the events? This conclusion should be more clearly explained or removed. Furthermore, the preceding sentence requires grammatical revision, specifically "… are failed to be identified."**

*AR: The text has been revised accordingly, the highlighted sentece has been removed and the grammar of the preceding sentence adjusted.*

**Technical Corrections:**

**RC: Line 24: Though understood, the phrase "… and so its water holding capacity;" reads awkwardly in this context.**

**RC: Line 33: The phrase "… makes its observation still a major challenge…" probably can drop the word "*still*."**

**RC: Line 34: Similarly, the sentence with "… hail observing system is still missing" could benefit from a revision that does not include the word "*still*."**

**RC: Lines 35 and 86: Change "including" to "that include", otherwise it reads awkwardly.**

**RC: Line 43: The phrase "… constitute a precious way…" is distracting. Consider rephrasing and without use of the word "*precious*".**

**RC: Line 73: "These are…" – What are? No specific parameter was mentioned yet. Again, this may just be a structure issue. Consider revising this sentence and the one preceding.**

**RC: Line 88: Sentence requires grammatical revision.**

**RC: Line 94 and 98: What does "description" mean in these contexts? As in "characterization" or "depiction"?**

**RC: Lines 146 and 454: Suggest revision of the phrase, "…actually occurred OTs…", e.g., "true OTs".**

**RC: Line 97: Consortium for Small-Scale Modelling?**

**RC: Line 185: "Their different formulations… distinct parts of the numerical model equations…" – perhaps, though arguable. The sentence isn't necessary – consider removing.**

**RC: Line 223: It is unclear if "latter study" is referring to the approach of Punge et al. – check use of "latter" in this context.**

**RC: Line 246: Perhaps "… filter mainly over *certain areas* of the sea…", because largely the pattern across sea to land is rather continuous, especially for a) and b).**

**RC: Lines 284 and 524: "Anyhow" should usually be avoided in formal writing. Furthermore, the structure of the Line 284 sentence, with too many qualifiers followed by an "empty *this*," makes it difficult to follow. Please revise.**

**RC: Lines 465-466: The ending of this sentence does not read easily. Consider revising.**

**RC: Line 470: Conclusions are typically separate from Discussion.**

*AR: We thank the reviewer for the technical corrections and suggestions, all the proposed modifications have been accepted and the text revised accordingly.*

*AR: References*

*Gagne, D. J., McGovern, A., Haupt, S. E., Sobash, R. A., Williams, J. K., & Xue, M. (2017). Storm-based probabilistic hail forecasting with machine learning applied to convection-allowing ensembles. Weather and forecasting, 32(5), 1819-1840.*

*Gensini, V. A., Converse, C., Ashley, W. S., & Taszarek, M. (2021). Machine learning classification of significant tornadoes and hail in the United States using ERA5 proximity soundings. Weather and Forecasting, 36(6), 2143-2160.*

*Prein, A. F., & Holland, G. J. (2018). Global estimates of damaging hail hazard. Weather and Climate Extremes, 22, 10-23.*

*Púčik, T., Castellano, C., Groenemeijer, P., Kühne, T., Rädler, A. T., Antonescu, B., & Faust, E. (2019). Large hail incidence and its economic and societal impacts across Europe. Monthly Weather Review, 147(11), 3901-3916.*

*Scarino, B., Itterly, K., Bedka, K., Homeyer, C. R., Allen, J., Bang, S., & Cecil, D. (2023). Deriving Severe Hail Likelihood from Satellite Observations and Model Reanalysis Parameters Using a Deep Neural Network. Artificial Intelligence for the Earth Systems, 2(4), 220042.*

*Torralba, V., Hénin, R., Cantelli, A., Scoccimarro, E., Materia, S., Manzato, A., & Gualdi, S. (2023). Modelling hail hazard over Italy with ERA5 large-scale variables. Weather and Climate Extremes, 39, 100535.*

---

## Author Comment (AC2)

**Authors replies to reviewer comments RC2**

**Review for EGUSPHERE-2023-2639**

*Characterizing hail-prone environments using convection-permitting reanalysis and overshooting top detections over south-central Europe*

*Antonio Giordani, Michael Kunz, Kristopher M. Bedka, Heinz Jürgen Punge, Tiziana Paccagnella, Valentina Pavan, Ines M. L. Cerenzia, and Silvana Di Sabatino*

We want to thank the reviewer for his/her assessment of our manuscript. In the following we give our answers to the comments and recommendations that have been raised. Reviewer comments RC are **bold**, our replies AR are in *italic*.

**General comments:**

**RC: The study addresses the challenges associated with reliably observing and simulating hazardous hailstorms. The authors propose an approach that combines information from different sources, including remote sensing instruments, observations, and numerical modeling, to enhance the understanding of the spatial and temporal patterns of severe hail occurrences in south-central Europe. The methodology involves developing a proxy for hail frequency by integrating overshooting cloud top (OT) detections from the Meteosat Second Generation (MSG) weather satellite with convection-permitting SPHERA reanalysis predictors describing hail-favorable environmental conditions.**

**While the paper is already quite robust, there are a few shortcomings which should be addressed by the authors to enhance the readability and importance of their work.**

**I recommend major revision with consideration of the specific comments listed below.**

**Specific comments:**

**Section 2**

**RC: While the methodology is generally well-described, ensuring greater clarity would enhance the paper's accessibility. Providing, for example, a flowchart for data processing and analysis could be beneficial.**

*AR: We thank the reviewer for the very nice suggestion. In the revised manuscript a flowchart has been added (reported below) which summarizes the analyses contained in the manuscript. We hope that this could enhance the readability and clarity of the paper.*

[Figure]

**Section 2.2**

**RC: I am wondering if you considered using the 5-minute rapid scan data from MSG, which may improve the quality of this paper, since you state in lines 538-544 that the low temporal resolution is a limitation of your work.**

*AR: We agree with the reviewer that the usage of 5-minute rapid cloud-top scans from the MSG could have enhanced the efficacy and significance of the work. Unfortunately, at the time of the analysis those higher-resolution data were not easily accessible in the necessary format (McIDAS AREA) via the Data Center of the University of Wisconsin Space Science and Engineering Center needed to run the Khlopenkov et al., 2021 algorithm for automatic OT detection. Further, ordering many years of rapid scan data from the EUMETSAT archive is a very time consuming and inefficient process, so this was a further limiting factor. Hence, the best available option was to rely on MSG scans data with a frequency update of 15 minutes. In the revised manuscript an explicit statement has been added to justify this choice.*

**Section 2.3:**

**RC: Why hasn't the low-level moisture been added to the filter? You clearly and rightfully state that it is an important factor for hail formation, so I don't really understand why the moisture wasn't considered for the filter.**

*AR: The amount of low-level moisture is implicitly included in the parameters entering the filter conditionally on the presence of instability as in the formulation of the K index whose 2nd and 3rd additive components (i.e., $Td_{850}$, dew-point temperature at 850 hPa, and ($T_{700} - Td_{700}$), difference between the ambient temperature and the dew-point temperature at 750 hPa) quantify respectively the low-level moisture content and the moist layer depth (as reported in Appendix A). The inclusion of the low-level moisture within the K index has been highlighted multiple times within the text (e.g. lines: 175/187/248/434). However, in the revised manuscript we attempted to be even more explicit and added a statement in Section 2.3 to make it clearer.*

**RC: Moreover, a recent study showed that CAPE above the -10°C-isotherm stood out as the best predictor for Europe (Battaglioli et al. 2023). Another recent study (Nixon et al. 2023), for the US this time, showed that the depth of the storm ("maximum parcel level") and storm-relative winds below the hail-growth-layer may play a key role in formation and size of hailstones. Hence, I would suggest rethinking the choice of ambient predictors.**

*AR: We thank the reviewer for the suggestions. As of today, there is not general consensus in the scientific community on which are the best thermodynamical predictors specifically to describe convective environments supporting hail development. The choice on the environmental predictors included in this work relied upon those parameters that are mostly used for the purpose in central-European regions, based on previous findings that mainly inspired this work (Punge et al., 2017), as well as literature that reported their quantitative added value over the common investigated area (Kunz 2007, Kunz et al., 2020, Jelic et al., 2020). Further, the referenced work (Battaglioli et al., 2023) was unpublished at the time of the proceedings of the present work. We would also like to point out that the filter approach proposed is not foreseen for positive hail predictions, nor for the definition of a robust climatology of potential hail events, but is mainly designed to remove non-hail occurrences based on the identification of the minimum environmental conditions necessary for supporting hail development. Hence, we believe that the chosen set of predictors is adequate for this purpose, owing also to the conservative nature of the filter designed (which ambient thresholds are defined to remove all those occasions which are unlikely to have produced hail by excluding a portion relative to the 5% of the whole single parameters distributions), and which could hardly benefit from a change of e.g. the formulation of the CAPE parameter. Furthermore, extracting the reanalysis fields including the information on this alternative formulation of the CAPE parameter is not a straightforward task and would require significant effort. Additionally, the potential inclusion of the the storm-relative winds below the hail-growth-layer, despite being very promising as demonstrated by recent research (e.g., Kumjan & Lombardo, 2020), we believe to be significantly complicated due to the necessary inclusion of the information on the hail-growth zone of the storm which is still very challenging to estimate, and is not possible to account for with the reanalysis data included in this work. That said, it is certainly interesting for possible future extension of the work to include other different parameters such as those proposed by the reviewer. In the revised manuscript we expanded the future outlook of the Conclusion section including that possibile future extension of the work could focus more on the analysis with additional environmental parameters to improve the identification of hail-producing environments*

**Section 5:**

**RC: I would suggest splitting the discussion and conclusion. The current content of this section is not well structured, and the take-home messages are not clearly outlined.**

*AR: We thank and agree with the reviewer. In the revised manuscript the Discussion (Section 5) is clearly separated from the Conclusion (Section 6).*

**RC: Providing context for the practical applications of the research would also highlight the significance of this paper (e.g., risk assessment, insurance).**

*AR: We appreciate the suggestion of the reviewer. In the revised manuscript we added a brief discussion in the future outlook of the Conclusion section highlighting potential downstream*

*applications of the presented analysis (such as its inclusion in risk assessment strategies or for insurance purposes) with the aim to enhance the relevance of this scientific work.*

**Figure 1:**

**RC: Large hail is defined with "≥ 2 cm" in the ESWD (and not "≥ 3 cm"). So, this error needs to be addressed throughout the manuscript to maintain coherence.**

*AR: We thank the reviewer for pointing out this mismatch between our analysis and the ESWD nomenclature. However, since the adjective "large" is arbitrary on the definition choice, and since the paper is self-contained and completely independent from the ESWD database, we do not believe that we should rename the hail size characterization used in the manuscript. Furthermore, as reported in the review of Raupach et al., 2021, the exact hailstone diameter defining "large" or "very large" hail is a matter of definition and varies in the literature, and their proposal is to refer to severe hail as that with hailstones of at least 2 cm in diameter, large hail as that with at least 3.5 cm diameter hailstones and very large hail as that with hailstones of at least 5 cm diameter. Anyhow, to enhance clarity and avoid possible misinterpretations, an explicit statment has been added at the beginning of Section 2.1 to point out the different nomenclature between the paper and ESWD.*

**Technical corrections:**

**RC: Line 203: "intrinsic"**

**RC: Line 519: "large hail-producing storms"**

**References:**

Battaglioli et al. 2023 (https://doi.org/10.1175/JAMC-D-22-0195.1)

Nixon et al. 2023 (https://doi.org/10.1175/WAF-D-23-0031.1)

*AR: References*

*Khlopenkov, K. V., Bedka, K. M., Cooney, J. W., & Itterly, K. (2021). Recent advances in detection of overshooting cloud tops from longwave infrared satellite imagery. Journal of Geophysical Research: Atmospheres, 126(14), e2020JD034359.*

*Kumjian, M. R., & Lombardo, K. (2020). A hail growth trajectory model for exploring the environmental controls on hail size: Model physics and idealized tests. Journal of the Atmospheric Sciences, 77(8), 2765-2791.*

*Kunz, M. (2007). The skill of convective parameters and indices to predict isolated and severe thunderstorms. Natural Hazards and Earth System Sciences, 7(2), 327-342.*

*Kunz, M., Wandel, J., Fluck, E., Baumstark, S., Mohr, S., & Schemm, S. (2020). Ambient conditions prevailing during hail events in central Europe. Natural Hazards and Earth System Sciences, 20(6), 1867-1887.*

Jelić, D., Megyeri, O. A., Malečić, B., Belušić Vozila, A., Strelec Mahović, N., & Telišman Prtenjak, M. (2020). Hail climatology along the northeastern Adriatic. *Journal of Geophysical Research: Atmospheres, 125(23)*, e2020JD032749.

Punge, H. J., Bedka, K. M., Kunz, M., & Reinbold, A. (2017). Hail frequency estimation across Europe based on a combination of overshooting top detections and the ERA-INTERIM reanalysis. *Atmospheric Research, 198*, 34-43.

Raupach, T. H., Martius, O., Allen, J. T., Kunz, M., Lasher-Trapp, S., Mohr, S., ... & Zhang, Q. (2021). The effects of climate change on hailstorms. *Nature reviews earth & environment, 2(3)*, 213-226.